# Separation of scales and a thermodynamic description of feature learning in some CNNs

Inbar Seroussi [1] ✉, Gadi Naveh[2] & Zohar Ringel[2]

Deep neural networks (DNNs) are powerful tools for compressing and distilling information. Their scale and complexity, often involving billions of interdependent parameters, render direct microscopic analysis difficult. Under such circumstances, a common strategy is to identify slow variables that average the erratic behavior of the fast microscopic variables. Here, we identify a similar separation of scales occurring in fully trained finitely over-parameterized deep convolutional neural networks (CNNs) and fully connected networks (FCNs). Specifically, we show that DNN layers couple only through the second cumulant (kernels) of their activations and pre-activations. Moreover, the latter fluctuates in a nearly Gaussian manner. For infinite width DNNs, these kernels are inert, while for finite ones they adapt to the data and yield a tractable data-aware Gaussian Process. The resulting thermodynamic theory of deep learning yields accurate predictions in various settings. In addition, it provides new ways of analyzing and understanding DNNs in general.

Identifying slow or relevant variables is an essential step in analyzing large-scale non-linear systems. In the context of deep neural networks (DNNs), these should be some combinations of the individual weights that are weakly fluctuating and obey a closed set of equations. One potential set of such variables is the DNNs' outputs themselves. Indeed, in the limit of infinitely over-parameterized DNNs these provide an elegant picture of deep learning[1–3] based on a mapping to Gaussian Processes (GPs). However, these GP limits miss out on several qualitative aspects, such as feature learning[4,5] and the fact that real-world DNNs are not nearly as over-parameterized as required for the GP description to hold[1,3,6,7]. Obtaining a useful set of slow variables for describing deep learning at finite over-parameterization is thus an important open problem in the field.

Several works provide guidelines for this search. Noting that GP limits can have surprisingly good performance[8] and that over-parameterization is natural to deep learning[9,10] we are inclined to keep some elements of the GP picture. One such element is to work in function space and study pre-activation and outputs instead of weights whose posterior distribution becomes complicated even in the GP limit[11,12]. Another element is the layer-wise composition of hidden layer kernels[13] which yields the output kernel of the GP[14]. Such a layer-wise picture is also harmonious with the idea that DNN layers should not correlate strongly, to prevent co-adaptation[15]. Recently, it was shown that in some limited settings, making the GP kernel "dynamical" or flexible, so that it adapts to the dataset, can account for differences between infinite and finite DNNs[3,16–19]. Still, the task of finding an explicit set of equations describing this flexibility in deep non-linear DNNs remains unsolved. Specifically, while in the GP limit we find tractable algebraic expressions, involving only basic matrix manipulations, for the DNN's prediction in the feature learning regime similar expressions only exist for deep linear[16,20] or non-linear networks with one trainable layer[21–23]. In a related manner, while the DNN's outputs provide a complete set of slow variables in the GP limit, in the finite-width feature learning regime it is not clear which subset of variables governs the trained DNN's behavior other than the entire set of weights.

In this work, we identify such slow variables and use these to derive an effective theory for deep learning capable of capturing various finite channel/width (*C/N*) effects (such as feature learning) in convolutional neural networks (CNNs) and fully connected neural networks (FCNs). We argue that:

[1]Weizmann Institute of Science, Department of Mathematics, Rehovot 7610001, Israel. [2]Hebrew University, Racah Institute of Physics, Jerusalem 9190401, Israel. ✉ e-mail: inbarser@gmail.com

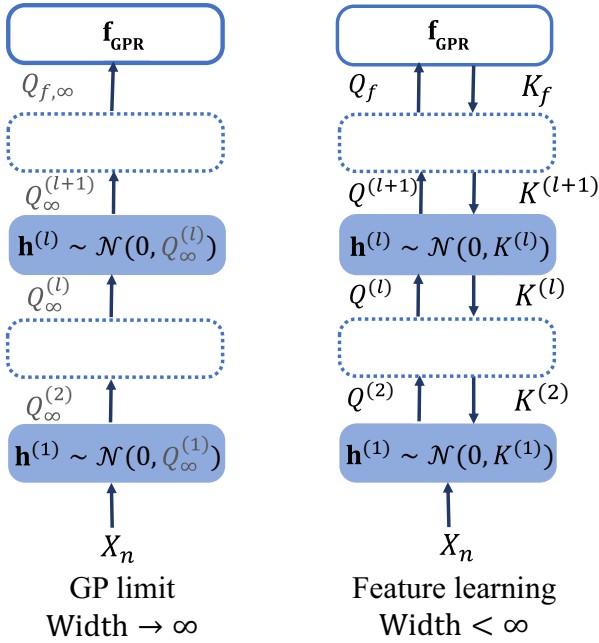

**GP limit**
Width $\rightarrow \infty$

**Feature learning**
Width $< \infty$

**Fig. 1 | Feature learning regime versus Gaussian process infinite limit.** Learning as described by our effective theory for antisymmetric activation functions. Left: for infinite width the pre-activations ($h^{(l)}$), and the output ($f$) fluctuate according to a Gaussian distribution with fixed post-kernels $Q_\infty^{(l)}$, $Q_{f,\infty}$, respectively. The complete set of slow variables are the outputs with fixed kernels. *Right:* for large but finite width and number of samples, we obtain an approximately Gaussian distribution for the pre-activations and outputs with learned pre-kernels $K^{(l)}$ and $K_f = Q_f + \sigma^2 I_n$, respectively. The outputs follow a Gaussian Process Regression (GPR) with kernel $Q_f$. The complete set of slow variables here are both the outputs and the pre and post-kernels. For general activation, an additional slow variable, corresponding to the mean of the preactivation, needs to be tracked.

1. For, $C, N \gg 1$ the erratic behavior of specific channels/neurons averages out and hidden layers coupled to each other only through two "slow" variables per layer: the second cumulant of the pre-activations (pre-kernel), $K^{(l)}$, and the second cumulant of activations (post-kernel), $Q^{(l)}$, of the *l*th layer. Furthermore, for mean square error (MSE) loss, FCNs in the so-called *mean-field (MF) scaling* (where the last layer weights are scaled down) or CNNs with a large read-out layer fan-in behave effectively as a GP with a data-aware kernel determined by the second cumulant of pre-activations in the penultimate layer[21]. For comparison between the classical GP limit and the GP process we find, see Fig. 1.

2. In settings where the kernels have a large density of dominant eigenvalues, the posterior (or trained) pre-activations fluctuate in a nearly Gaussian manner. Following this, we use a multivariate Gaussian variational approximation for the posterior pre-activations and derive explicit matrix equations (Equations of State) for the covariance matrices governing these pre-activations and the DNNs predictions.

3. We identify an emergent feature learning scale (FLS) denoted by $\chi$, proportional to the train MSE times $n^2$ over $C$ (or $N$). This scale controls the difference between the finite $C, N$ output kernel ($Q_f$) and its $C, N \rightarrow \infty$ limit and in this sense reflects feature learning. Due to the $n^2$ factor, $\chi$ can be $O(1)$ or larger even for $C \gg 1$, e.g., for CNN architectures (see Fig. 2 panel c). The same holds, with $C$ replaced by $N$, for FCNs in the MF scaling[21]. Unlike perturbation theory[3,6,23,24], our theory tracks all orders of $\chi$ and treats only $1/C, 1/N$ perturbatively. The separation of scales between $\chi$ and $1/C, 1/N$ is

thus central to our analysis. Its manifestation is the fact that feature learning shifts and stretches the dynamical variables in the theory (the pre-activations) in a considerable manner yet barely spoils their Gaussianity.

4. We provide what is, to the best of our knowledge, the first analytic predictions for the test performance of a non-linear network with two trainable layers in the feature learning regime. We do so both for an FCN and a CNN.

The predictions of our approach are tested on several toy and real-world examples using direct analytical approaches and numerical solutions to the equations of state. Our analysis takes a physics viewpoint on this complex non-linear problem. Rigorous mathematical proofs are left as an open problem for future research.

We note that there are several works showing evidence that the spectrum of the empirical weight correlation matrix show various tail effects and spikes[25]. While in deeper layers we focus on pre-activations, the spectrum of input layer weights we obtained, is Gaussian but not independent as in ref. [12]. Hence, it can produce a variety of spectral distributions for the covariance matrix, similar to the aforementioned ones. We note a recent interesting work[26] arguing that the test-loss depends only on the mean and variance of hidden activations. There, however, the setting is of a fixed trained DNN and the statistics are over the input measure rather than over the DNN parameters as in our case. While quantitatively different, our approach is similar in spirit to the layer-wise Gaussian Processes algorithm[19] inspired by DNN experiments. However, our approach provides a more accurate first principles description of trained DNNs. Additional approaches for finite-width include perturbative correction around the infinite width limit to leading[3,24] or higher orders[27–29]. There is however mounting evidence from bounds on GP limits[1,28], numerical experiments[3,5,6,23], as well as the current work, that such perturbative expansions have slow convergence in practical regimes. In contrast, our EoS are useful both numerically (see section "Methods" and "Numerical demonstration: 3 layer FCN") and analytically (see section "Analytical solution of the EoS - two layer CNN") and in addition allow us to model pre-activation distributions in the wild via our pre-kernels (see section "Extensions to deeper CNNs and subsets of real-world data-sets").

## Results
### Problem statement

Our general setup consists of DNNs trained on a labeled training set of size $n, \mathscr{D}_n = \{(\mathbf{x}_\mu, y_\mu)\}_{\mu=1}^n = \{X_n, \mathbf{y}\}$ with MSE loss. The input vector is $\mathbf{x}_\mu \in \mathbb{R}^d$, and the target $y_\mu$ is a scalar output. We denote vectors and tensors by boldface and use $\mu, \nu$ to represent data point indices.

Our theory can be applied to any finite number of convolutional, dense, or pooling layers. To illustrate its main aspects, let us focus on an $L$-layer fully connected model with width $N_l$ ($l \in [1..L-1]$),

$$f_{\boldsymbol{\theta}}(\mathbf{x}) = \sum_{j=1}^{N_L} w_j^{(L)} \phi\left(h_j^{(L-1)}(\mathbf{x})\right); \; h_j^{(l)}(\mathbf{x}) = \sum_{i=1}^{N_{l-1}} W_{ji}^{(l)} \phi\left(h_i^{(l-1)}(\mathbf{x})\right); \quad \mathbf{h}^{(1)}(\mathbf{x}) = W^{(1)}\mathbf{x}$$

(1)

where $\boldsymbol{\theta} := \text{vec}\{W^{(1)}, \ldots, W^{(L-1)}, w^{(L)}\}$ are the trainable parameters of the network arranged in a vector such that ($i \in [1, N_1]$), $\mathbf{w}^{(L)} \in \mathbb{R}^{N_{L-1}}$, $W^{(l)} \in \mathbb{R}^{N_l \times N_{l-1}}$ are the weights of the network, and the input vector $\mathbf{x} \in \mathbb{R}^d$ such that $d = N_0$. The activation function, $\phi : \mathbb{R} \rightarrow \mathbb{R}$, is applied element-wise.

We take the Bayesian Neural network perspective, i.e. the above network is a random object drawn from a set of Neural networks. The main object we analyze is the Bayesian posterior distribution of the DNN outputs $p(\mathbf{f}|\mathscr{D}_n)$. The target $y_\mu$ is the output of the network $f_{\boldsymbol{\theta}}(\mathbf{x}_\mu)$ with additive i.i.d centered Gaussian noise with variance $\sigma^2$ and an i.i.d Gaussian prior on weights $W_{ij}^{(l)} \sim \mathcal{N}(0, \sigma_l^2/N_l)$. The posterior can be rewritten as $p(\mathbf{f}|\mathscr{D}_n) \propto p(\mathbf{f}|X_n) \exp\left(-\sum_\mu (y_\mu - f_{\boldsymbol{\theta}}(\mathbf{x}_\mu))^2 / 2\sigma^2\right)$ where the

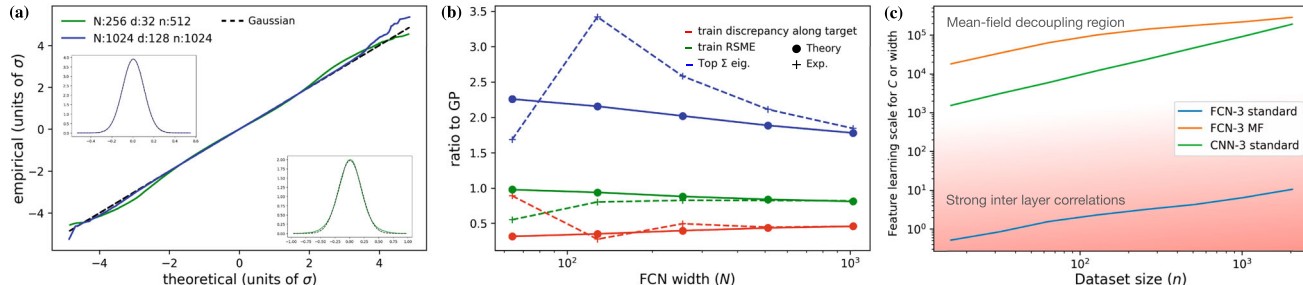

**Fig. 2 | Theory versus experiment. a, b** Compare fully trained 3-layer FCNs in the MF scaling with our theoretical predictions. **a** Studies the most strongly fluctuating weight mode in the first layer. A QQ-plot is shown, of its empirical distribution against a normal one, along with histograms together with Gaussian fits (insets). As the scale of the problem increases, these fluctuations become more and more Gaussian. **b** The experimental values and EoS predictions, each normalized by their values at $N_l \to \infty$ with fixed $\sigma_l$ (i.e. the GP value). Here we used $n = 1024$ and $d = 128$. **c** Plots the $N$ (or channel number $C$) at which the feature learning scale ($\chi$) is 1. This

dictates the crossover between weak and strong feature learning in our theory. The color gradient provides a qualitative separation between large and small, $N, C$ and thus correlates with the adequateness of our mean-field decoupling for the hidden layers. Evidently, FCNs with standard scaling ($[Q_f]_{\mu\mu} = O(1)$ at $n = 0$) require $N_l = O(1)$ for appreciable feature learning. However, FCN with an MF scaling, as well as CNNs in standard scaling, can exhibit strong feature learning well within the regime of our mean-field decoupling.

prior distribution is

$$p(\mathbf{f}|X_n) = \left\langle \prod_\mu \delta\left[f(\mathbf{x}_\mu) - f_\theta(\mathbf{x}_\mu)\right] \right\rangle_\theta, \qquad (2)$$

where $\langle \dots \rangle_\theta$ denotes an average over the prior distribution of the weights. We note that there is a correspondence between certain gradient-based methods and Bayesian Neural Networks (BNNs)[30–32]. Recent work[33] shows that BNNs can perform on par or even better than DNNs trained with SGD. In particular, the equilibrium distribution of the Langevin-type training algorithm, i.e., full-batch gradient descent with a small learning rate, along with weight decay, and additive white noise with variance $\sigma^2$[23] is equivalent to the posterior distribution of a BNNs, $p(\mathbf{f}|\mathscr{D}_n)$ is in function space. Adopting physics notation, it can be written as $p(\mathbf{f}|\mathscr{D}_n) = e^{-\mathscr{S}}/\mathscr{Z}(\mathscr{D}_n)$, where $\mathscr{Z}(\mathscr{D}_n) = \int e^{-\mathscr{S}}$ is the partition function, and $\mathscr{S}$ is the action or negative log-posterior (see also Methods). As shown in Supplementary Material (1) this action is given by

$$\mathscr{S} = \frac{1}{2}\sum_{i=1}^{N_1}(\mathbf{h}_i^{(1)})^\top \left[Q^{(1)}\right]^{-1}\mathbf{h}_i^{(1)} + \frac{1}{2}\sum_{l=2}^{L-1}\sum_{i=1}^{N_l}(\mathbf{h}_i^{(l)})^\top\left[\tilde{Q}^{(l)}(\mathbf{h}^{(l-1)})\right]^{-1}\mathbf{h}_i^{(l)}$$
$$+ \frac{1}{2}\mathbf{f}^\top\left[\tilde{Q}_f(\mathbf{h}^{(l-1)})\right]^{-1}\mathbf{f} + \frac{1}{2\sigma^2}\sum_\mu(f_\mu - y_\mu)^2 \qquad (3)$$

where $f_\mu = f(\mathbf{x}_\mu)$, $h_{i\mu}^{(l)} = h_i^{(l)}(\mathbf{x}_\mu)$ and

$$Q_{\mu\nu}^{(1)} = \frac{\sigma_1^2}{d}\mathbf{x}_\mu^T\mathbf{x}_\nu, \quad \tilde{Q}^{(l+1)}(\mathbf{h}^{(l)})_{\mu\nu} = \frac{\sigma_{l+1}^2}{N_l}\sum_{i=1}^{N_l}\phi(h_{i\mu}^{(l)})\phi(h_{i\nu}^{(l)}),$$
$$\tilde{Q}_f(\mathbf{h}^{(L-1)})_{\mu\nu} = \frac{\sigma_L^2}{N_{L-1}}\sum_{j=1}^{N_{L-1}}\phi(h_{j\mu}^{(L-1)})\phi(h_{j\nu}^{(L-1)}) \qquad (4)$$

where $\mathbf{h}_i^{(l)} \in \mathbb{R}^n$. We comment that for rank-deficient matrices, the inverses are regularized by including a small positive regularizer to be taken to zero at the end of the computation.

To familiarize ourselves with the action in Eq. (3) let us see how it reproduces the standard Gaussian Processes picture at infinite channel/width[3,14]. Strictly speaking, this action is highly non-linear, since the $\tilde{Q}$'s matrix elements contain high powers of pre-activations and since their inverse enters the action. Crucially, however, the $\tilde{Q}$'s are width-averaged quantities. Thus, at $N_l \to \infty$ one may replace them by their averages. Furthermore, upstream dependencies, wherein $\mathbf{h}^{(l)}$ affects $\mathbf{h}^{(l-1)}$, vanish (see Supplementary Material (1)). Roughly speaking, this is because $\mathbf{h}^{(l)}$ only feels the collective effect of all the

neurons in $\mathbf{h}^{(l-1)}$ rendering its feedback on any specific neuron negligible.

Having these two simplifications in mind, we begin a layer-by-layer downstream analysis of the DNN: As there is no upstream feedback on $\mathbf{h}^{(1)}$, the average of $\tilde{Q}^{(2)}(\mathbf{h}^{(1)})$ (denoted $Q^{(2)}$) can be carried under the Gaussian action of the input layer alone (first term in the action). Replacing $\tilde{Q}^{(2)}(\mathbf{h}^{(1)})$ by $Q^{(2)}$ in the second term in the action, would then imply that $\mathbf{h}^{(2)}$ fluctuates in a Gaussian manner with $Q^{(2)}$ as its covariance matrix. Next, the average of $\tilde{Q}^{(3)}(\mathbf{h}^{(2)})$ (denoted $Q^{(3)}$) can be found based on the now known, Gaussian statistics of $\mathbf{h}^{(2)}$. Repeating this process, the final kernel ($Q^{(L)} = Q_f$) is found and is exactly the one obtained using the method introduced by Cho and Saul[14]. Together with the MSE loss term (last term in the action), we find that the outputs ($f_\mu$) fluctuate in a Gaussian manner, leading to the standard GP picture of infinite width trained DNNs[3].

Here, however, our focus is at large but finite width ($N_l \gg 1$). In this more complex regime, several corrections may appear: (i) The pre-activations' average and covariance may deviate from those of a random DNN. (ii) $Q^{(l)}$, the covariance of activations in the $l-1$ layer, would not solely determine the covariance of pre-activations in the downstream layer $l$, as upstream effects between $\mathbf{h}^{(l+1)}$ and $\mathbf{h}^{(l)}$ come into play. (iii) Inter-channel (or inter-neuron in the fully connected case) and inter-layer correlations may appear. (iv) The fluctuations of pre-activations may deviate from that of a Gaussian. A priori, all these corrections may play similarly dominant roles, thereby making analysis cumbersome.

### Effective GP description in the feature learning regime

The basic analytical insight underlying this work is that these four types of corrections scale differently with $n$, $d$, and $N_l$. This allows for a controlled mean-field treatment, which differs substantially from straightforward perturbation theory in one over the width. As shown in Supplementary Material (1.6), corrections of type (iii) are often much smaller than those of type (i) and (ii). This holds generally for hidden layers when $N_l$'s (or channel's number for CNNs) are much larger than 1. Considering the output layer of CNNs this requires a large fan-in and FCNs this holds when using an MF scaling[21]. Turning to correction of type (iv) in the $l'$th layer, these are suppressed when the average $\tilde{Q}^{(l)}$ has a large density of dominant eigenvalues—a situation relevant for when $n$ and the input dimension are both large relative to one.

This leaves us with corrections of types (i) and (ii). Interestingly, following these corrections to all orders leads to a tractable mean-field picture of learning. The latter is an augmentation of the standard correspondence between GPs and DNNs at infinite width (NNGP)[14,34]: Pre-activations in different layers or channels/neurons remain

uncorrelated and Gaussian. Correlations only appear between different data-points (and latent pixels for CNNs) within the same layer and channel/neuron. We henceforth denote the covariance of pre-activations and activations at layer $l$ (up to normalization by the variance of the weights) by $K^{(l)}$ and $Q^{(l)}$ and refer to these as pre-kernel and post-kernel, respectively. However, in the NNGP[34] viewpoint, $Q^{(l)}$ is simply proportional to $K^{(l)}$ and fully determined by the upstream kernel ($Q^{(l-1)}$) whereas here $K^{(l)}$ and $Q^{(l)}$ differ and moreover depend both on the upstream and downstream kernels. Below, we present our theory for antisymmetric activation function, generalizations to other activation function such as ReLU, where one needs to track both kernels and means of pre-activations, could be found in Supplementary Material (5). Specifically, for the above $L$-layers FCN with $\phi$ = erf activation function, we obtain (see also Fig. 7 and Supplementary Material (1) for derivation and extension to CNNs and any number of layers)

$$\bar{f} = Q_f \left[ Q_f + \sigma^2 I_n \right]^{-1} \mathbf{y}$$

$$\left[ \left( K^{(l-1)} \right)^{-1} \right]_{\mu\nu} = \left[ \left( Q^{(l-1)} \right)^{-1} \right]_{\mu\nu} - \frac{1}{N_{L-1}} \mathrm{Tr} \left\{ A^{(L)} \frac{\partial Q_f}{\partial \left[ K^{(l-1)} \right]_{\mu\nu}} \right\}$$

$$\left[ \left[ K^{(l-1)} \right]^{-1} \right]_{\mu\nu} = \left[ \left[ Q^{(l)} \right]^{-1} \right]_{\mu\nu} + \frac{2N_l}{N_{L-1}} \frac{\partial D_{\mathrm{KL}}(K^{(l)} \| Q^{(l)})}{\partial \left[ K^{(l-1)} \right]_{\mu\nu}} \quad \text{for all } l \in [2, L-1] \quad (5)$$

$$\left[ \Sigma^{-1} \right]_{ss'} = \frac{d}{\sigma_1^2} \delta_{ss'} + \frac{2N_2}{N_1} \frac{\partial D_{\mathrm{KL}}(K^{(2)} \| Q^{(2)})}{\partial \Sigma_{ss'}}$$

$$A^{(L)} = \sigma^{-4} (\mathbf{y} - \bar{\mathbf{f}})(\mathbf{y} - \bar{\mathbf{f}})^\top - \left[ Q_f + \sigma^2 I_n \right]^{-1}$$

where $[Q_f]_{\mu\nu} = \sigma_L^2 G(K^{(L-1)})_{\mu\nu}$, $[Q^{(l)}]_{\mu\nu} = \sigma_l^2 G(K^{(l-1)})_{\mu\nu}$ and $G(K)_{\mu\nu} = \frac{2}{\pi} \sin^{-1} \left( \frac{2K_{\mu\nu}}{\sqrt{1+2K_{\mu\mu}}\sqrt{1+2K_{\nu\nu}}} \right)$[35] for a matrix $K \in \mathbb{R}^{n \times n}$ (equivalent expressions are known for several common activation functions, such as ReLU[14]). Also, $\bar{f}$ is the average DNN output and $D_{\mathrm{KL}}(K\|Q)$ is the KL-divergence between two Gaussians with covariance matrices $K$ and $Q$. The input layer post-kernel is $K^{(1)} = X_n \Sigma X_n^\top$, where $\Sigma$ is the covariance matrix of input layer weights.

As their lack of dependence on width suggests, the first equation together with the definitions of the post-kernels $Q^{(l)}, Q_f$ are already present in the strict GP limit ($N_l \to \infty$). They are, respectively, the GP inference formula and standard kernel recursive equations of random DNNs[14] with erf activation. The remaining equations are, to the best of our knowledge, novel and follow the changes to the pre-kernels and post-kernels at finite $N_l$. These could be solved analytically in some simple cases (see subsection "Analytical solution of the EoS - two layer CNN for the case of two-layer CNN"). We note that for non-antisymmetric activation, one will also need to track the mean of each layer's pre-activation (see Supplementary Material (5)).

To get a qualitative impression of their role, one can consider the case where the penultimate layer ($l = L - 1$) is linear, in which case $Q_f = \sigma_L^2 K^{(L-1)}$. Consequently, $\frac{\partial [Q_f]_{\mu'\nu'}}{\partial [K^{(L-1)}]_{\mu\nu}} = \sigma_L^2 \delta_{\mu\mu'} \delta_{\nu\nu'}$, (where $\delta_{\mu\nu}$, with double index refers here to the Kronecker delta) and thus the second equation simplifies to

$$\sigma_L^2 Q_f^{-1} = (Q^{(L-1)})^{-1} - \frac{\sigma_L^2}{N_{L-1}} \left( \varepsilon \varepsilon^\top - \left[ Q_f + \sigma^2 I_n \right]^{-1} \right), \quad (6)$$

where $\varepsilon = (\mathbf{y} - \bar{f})/\sigma$. We note in passing that even for a non-linear penultimate layer, a similar term will arise from the expansion of $Q_f$ in $K^{(L-1)}$ to linear order. From the above form, several insights can be drawn.

First, we argue that the above equation implies that the trained DNN is more susceptible to changes along $\varepsilon$ than the DNN at $N_{L-1} \to \infty$. Noting how $Q_f^{-1}$ enters the action (Eq. (3)), it controls the stiffness

associated with fluctuations in f. Hence, $Q_f^{-1}$ makes fluctuations in the direction of $\varepsilon$ more likely than they are according to $(Q^{(L-1)})^{-1}$. Since $\varepsilon$ measures the discrepancy in train predictions, this effect reduces the discrepancy by making the DNN more responsive in these directions than it is at $N_{L-1} \to \infty$. The second term, proportional to, $\left[ Q_f + \sigma^2 I_n \right]^{-1}$ amounts to a negligible reduction in fluctuations along eigenvectors of $Q_f$ corresponding to eigenvalues which are larger than $\sigma^2$.

Using Eq. (6) one can also identify the aforementioned emergent feature learning scale (or FLS) namely, $\chi = N_{L-1}^{-1} \varepsilon^\top Q^{(L-1)} \varepsilon$. This scale represents the magnitude of the leading term when one Taylor expands $Q_f$ in $1/N_{L-1}$. When $\chi = O(1)$ or larger there is a significant change in the eigenvalues of $Q_f$ compared to $Q^{(L-1)}$ which indicates feature learning. On the other hand, when this quantity is small, we are closer to the GP regime (see Supplementary Material (1.6)). To assess the scaling $\chi$, one can consider the common situation where $\varepsilon$ has some non-negligible overlap with dominant eigenvectors of $Q^{(L-1)}$ whose eigenvalues are on the scale $\lambda$. Here we find $\chi \approx \lambda \cdot \mathrm{MSE}/\sigma^2 \cdot n/N_{L-1}$, where MSE denotes the mean train MSE which enters here via $\|\varepsilon\|^2/n$. Due to its explicit $n$ dependency, and for $\lambda = O(n)$ at large $n$[36] $-\chi$ maybe $O(1)$ even at very large $N_{L-1}$ and/or when the average MSE is rather small.

Figure 2c shows the value of $N_{L-1}$ (or $C_{L-1}$) at which $\chi = 1$ (i.e. $N_{L-1}$ or $C_{L-1}$ at which feature learning becomes a dominant effect) as a function of $n$ for several DNNs we study. The scale separation, demonstrated there by the fact that $\chi$ can be $O(1)$ in regions where $1/N_l$'s is negligible, is central to our analytical approach.

This scale $\chi$ is also the reason that naive perturbation theory in $1/N_l$ fails at large $n$[3,6,23,27], as it treats $\chi$ and $O(1/N_{L-1})$ on the same footing, since they both have a single negative power of $N_{L-1}$. In contrast, our EoS treat the FLS non-perturbatively.

Last, we stress that the EoS provide us with a concrete, effective GP description for the entire DNN as well as its hidden layers. A priori one would expect that the normality of pre-activations, a large $C, N$ trait, will be lost at finite $C, N$. Yet, we find that pre-activations remain Gaussian and accommodate strong feature learning effects while maintaining accurate predictions. This unexpectedly simple behavior opens various reverse engineering possibilities, wherein one infers the effective kernels from experiments and uses their spectrum and eigenvectors to rationalize about the DNN (see also Fig. 5). On this note, we comment that extending the EoS to test points is straightforward by formally treating the test point as an additional training point with its own "noise" parameter and then sending it to infinity (see Supplementary Material (1.4.1.)). For concrete EoS expressions which include test-points, see section "Analytical solution of the EoS - two layer CNN".

**Numerical demonstration: 3 layer FCN**

Next, we test the agreement between the above results and statements and actual trained DNNs, starting from the 3-layer FCN defined in Eq. (1) with $L = 3$. We focus here on a student-teacher setting with $n = 512$ or 1024 training data points drawn from iid Gaussian distributions with unit variance along each input dimension. The target was generated by a randomly drawn teacher FCN of the same type, only with $N_1 = N_2 = 1$. The student was trained using an analog scaling to the MF scaling[21], wherein the output layer weights are scaled down by a factor of $1/\sqrt{N_l}$. Whereas for the CNNs discussed below, this choice of scaling was not required, for FCNs we found it necessary for getting any appreciable feature learning at $N_1 = N_2 = N \gg 1$ (Fig. 2c).

As described in the Methods section, we trained 20 FCNs using our Langevin algorithm until they reached equilibrium. We use these trained FCNs to calculate various average quantities under our partition function (Eq. (3)). Specifically, we focused on: (i) The normalized train-loss on the scale of $\sigma^2$, namely $\mathrm{MSE}/(\sigma^2 \sum_\mu y_\mu^2)$ (ii) The eigenvalues ($\lambda_i, i \in [1..d]$) of the average $\Sigma$, where we average over neurons, training seeds, and training time (the latter within the equilibrium region). (iii) The normalized overlap ($\alpha$) between the

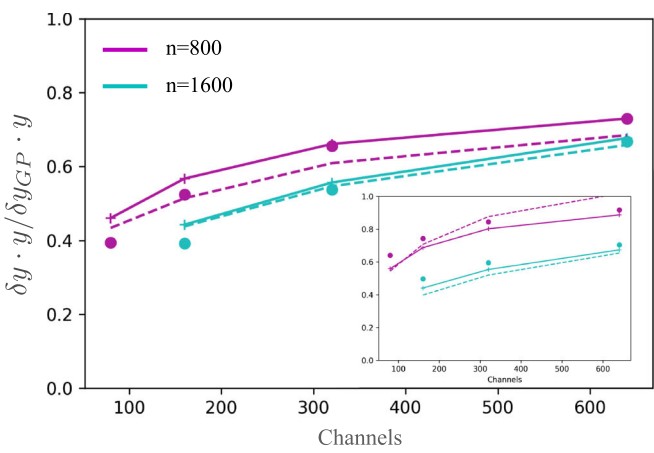

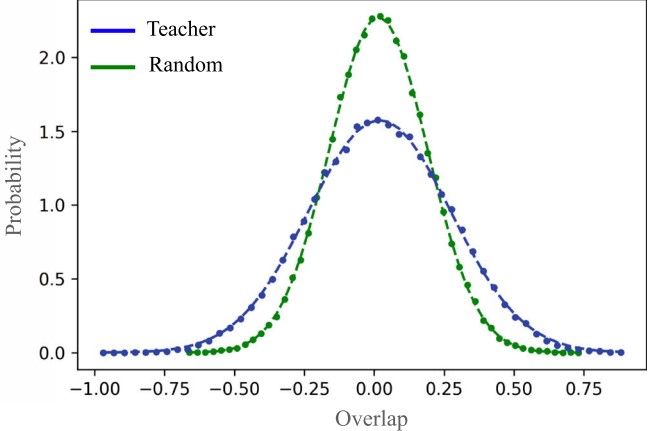

**Fig. 3 | Theory and experiment comparison for the 2-layer non-linear CNN.** Left. Discrepancy measured along target direction normalized by that of the corresponding GP ($\alpha/\alpha_{C\to\infty}$). Dots denote empirical values for the test-set. Dashed and solid lines are theoretical predictions via an approximate-analytical and exact-numerical solution of the equations of state, respectively. Train-set comparisons are shown in the inset. Right. Statistics of $\mathbf{w}\cdot\mathbf{w}^*/|\mathbf{w}^*|$ of the trained CNN (blue) compared with $\mathbf{w}$ projected on a normalized random vector (green) for the $n=1600$ experiment with $C=640$. Dashed lines show fit to a Gaussian.

discrepancy in prediction on the training set times the target, namely $\alpha = \sigma^{-2}\sum_{\mu=1}^{n}(\bar{f}_\mu - y_\mu)y_\mu/(\sum_\mu y_\mu^2)$. We then used a JAX-based[37] numerical solver for the EoS and compared it with the experiment.

As the results of Fig. 2b show, the predictions of our EoS for all these three quantities converged well as we increased $N$. Furthermore, they do so in a region where they differ considerably from their associated GP limit. Indeed, as shown in the Supplementary Material (3) the top $\Sigma$ eigenvalue came out 2–3 times larger than it is in the GP limit. The associated eigenvector corresponded to the first layer weights of the teacher ($\mathbf{w}^*$). The rest of the eigenvalues remained at their GP limit values. Put together, this is a clear sign of strong feature learning.

Notably, however, this notion of feature learning does not involve compression. Indeed, since $\Sigma$ has the same variance as in the GP limit for directions perpendicular to $\mathbf{w}^*$, it does not compress the input by projecting it solely on the label relevant direction ($\mathbf{w}^*$). Instead, it exaggerates the fluctuation of student weights along, $\mathbf{w}^*$ thereby making it statistically more likely that $h_{i\mu}^{(1)}$ and $h_{i\nu}^{(1)}$ with opposite sign of $\mathbf{w}^*\cdot\mathbf{x}_\mu$ and $\mathbf{w}^*\cdot\mathbf{x}_\nu$ will be further apart in the space of pre-activations.

Next, we study how $\chi$ behaves as a function of $n$ and $C$ (or $N$) for different architectures. Figure 2 shows the value of $N$ (or $C$) at which $\chi=1$. As $\chi$ contains a single inverse power of $C$ at ten times this value, $\chi$ would be 0.1 and thus indicate only minor feature learning effects in our EoS. As $N, C$ diminish from this latter value, our EoS yield increasingly stronger feature learning effects. We find that both for CNNs in the standard scaling and for FCNs with MF scaling, the crossover to feature learning happens well within the validity region of our mean-field decoupling (i.e. large $N$ or $C$). In contrast, FCN with standard scaling shows this crossover when $N=O(1)$, which is outside the scope of our theory. In this aspect, we comment that there is evidence that FCNs with standard scale are inferior to those with mean-field scaling[38] and perform similarly to GPs[34].

**Analytical solution of the EoS · two layer CNN**
Having tested our EoS numerically, we turn to show they lend themselves, in simple settings, to a fully analytical calculation. Amongst other things, this will flesh out the non-perturbative nature of our results. To this end, we consider a simple non-linear CNN with 2 layers. Though bounds have been derived[39,40], we are not aware of any analytical predictions for the performance of finite non-linear 2-layer DNNs, let alone CNNs. It is therefore a natural first application of our

approach. Specifically, we consider

$$f(\mathbf{x}) = \sum_{i=1}^{N}\sum_{c=1}^{C}a_{ic}\,\mathrm{erf}\left(\mathbf{w}_c\cdot\mathbf{x}_i\right) \tag{7}$$

where $\mathbf{x}\in\mathbb{R}^d$ with $d=NS$ and $\mathbf{w}_c,\mathbf{x}_i\in\mathbb{R}^S$. The vector $\mathbf{x}_i$ is given by the $iS,\ldots,(i+1)S-1$ coordinates of $\mathbf{x}$. The dataset consists of $\{\mathbf{x}_\mu\}_{\mu=1}^{n}$ i.i.d. samples, each sample $\mathbf{x}_\mu$ is a centered Gaussian vector with covariance $I_d$. We choose a linear target of the form $y_\mu = \sum_i a_i^*(\mathbf{w}^*\cdot\mathbf{x}_{\mu,i})$ where $a_i^* \sim \mathcal{N}(0,1/N)$ and $w_s^* \sim \mathcal{N}(0,1/S)$. This choice is not crucial, but does simplify considerably the GP inference part of the computation. We train this DNN using our Langevin algorithm and tune weight-decay and gradient noise such that, without any data, $a_{ic} \sim \mathcal{N}(0,\sigma_a^2(NC)^{-1})$ and $[w_c]_s \sim \mathcal{N}(0,\sigma_w^2 S^{-1})$.

The equations of state are given by (See Supplementary Material (3))

$$\bar{\mathbf{f}} = Q_f[\sigma^2 I_n + Q_f]^{-1}\mathbf{y}$$
$$\left[Q_f\right]_{\mu\nu} = \frac{\sigma_a^2}{N}\sum_i G\left(X_n\Sigma X_n^\top\right)_{\mu i,\nu i}$$
$$\left[\Sigma^{-1}\right]_{ss'} = \frac{S}{\sigma_w^2}\delta_{ss'} - \frac{1}{C}\mathrm{Tr}\left\{\left(\boldsymbol{\varepsilon}\boldsymbol{\varepsilon}^\top - K_f^{-1}\right)\frac{\partial Q_f}{\partial\Sigma_{ss'}}\right\} \tag{8}$$

Here we denote the discrepancy from the target by $\boldsymbol{\varepsilon}=(\mathbf{y}-\bar{\mathbf{f}})/\sigma^2$. The above equations for $\Sigma_{ss'}$ and $\delta_\mu$ could be solved numerically. Once $\Sigma_{ss'}$ is obtained, generalizing the EoS to test points is straightforward (see Supplementary Material (1.4.1)) and amounts to doing GP inference with $Q_f$. In particular, to obtain the required values of $Q_f$ between a test point $\mathbf{x}_*$ and a train point, $\mathbf{x}_\nu$ one calculates $2\sigma_a^2\sum_i\sin^{-1}\left(\frac{2x_{*,i}\Sigma x_{*,i}}{\sqrt{1+2x_{*,i}\Sigma x_{*,i}}\sqrt{1+2x_{\nu,i}\Sigma x_{\nu,i}}}\right)/(\pi N)$. The results for both train and test losses are shown in Fig. 3 in solid lines and match empirical values well.

To obtain fully analytical results, we proceed with several approximations for large $n$. First, we approximate the spectrum of the matrix $[Q_f]_{\mu\nu}$ based on its continuum kernel version $Q_f(\mathbf{x},\mathbf{x}')$. This is closely related to the equivalent kernel[41] approximation, which we adopt here along with its leading order correction[36]. Similarly, we use large $n$ to replace the double summation $\sum_{\nu\mu}\varepsilon_\mu\varepsilon_\nu[Q_f]_{\mu\nu}$ by two integrals over the measure from which $\mathbf{x}_\mu$ are drawn ($d\mu$). See

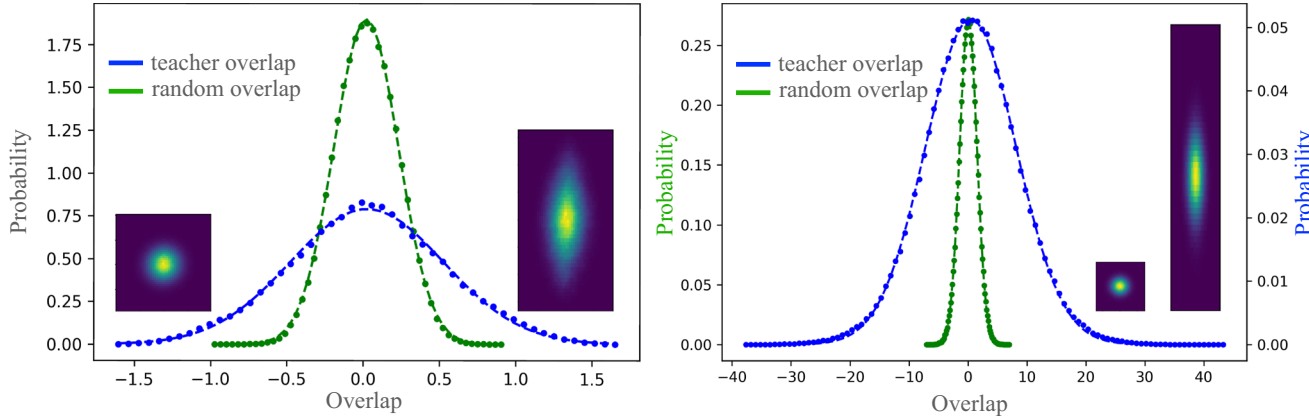

**Fig. 4 | Pre-activation statistics in the 3-layer student-teacher setting.** Left. Histogram of student input weight vector, dotted with the normalized teacher weight (blue) and normalized random weight (green). Right. Histogram of student hidden layer pre-activations, dotted with the normalized teacher pre-activations (blue) and normalized pre-activations of a random teacher (green). Dots are empirical values and dashed lines are Gaussian fits. Insets: 2d histograms along the same vectors before (left) and after (right) training. Within our framework, these variances are determined by $\mathbf{v}^{\top}K^{(l)}\mathbf{v}$ with $\mathbf{v}$ being either random unit vector or $h^{(l)}$ of the single channel teacher. Remarkably, despite strong changes to the kernels and various non-linearities in the action, the pre-activation is almost perfectly Gaussian.

Supplementary Material (3.1) for further details and a discussion of the fully connected case ($N = 1$).

Following our approximations, the equations acquire the full rotation symmetry of the data-set measure, which amount to an independent orthogonal transformation of each $x_i$. Furthermore, as shown in Supplementary Material (3), at large $S$, $\bar{f}(\mathbf{x})$ (the continuum function representing $f_\mu$) is linear given a linear target, $y(\mathbf{x})$, regardless of $\Sigma$ and hence so is $\epsilon(\mathbf{x})$. The above symmetry then implies that $\epsilon(\mathbf{x})$ is only a function of $\mathbf{w}^* \cdot \mathbf{x}_i$ and furthermore takes the simple form $\epsilon(\mathbf{x}) = \alpha y(\mathbf{x})$. The quantity $\alpha$ thus measures the overlap between the discrepancy in predictions ($\epsilon$) and the target. Following this, the EoS are reduced to a non-linear equation in a single variable $\alpha$

$$\sigma^2 \alpha = 1 - \frac{q_{\text{train}} S \lambda_\infty l_*}{S \lambda_\infty l_* + \sigma^2/n} \qquad (9)$$

where $l_* = [1 - \chi_2]^{-1}/S$, $\chi_2 = C^{-1} \alpha^2 n^2 \lambda_\infty$, and $\lambda_\infty$ is the dominant eigenvalue of $Q_f(\mathbf{x}, \mathbf{x}')$ associated with a linear function in the limit $C \to \infty$, $l_*$ is the eigenvalue of $\Sigma$ associated with $\mathbf{w}^*$ (the remaining eigenvalues are inert and equal to $1/S$), $\chi_2$ is the FLS for 2-layer CNN, and we assumed $||\mathbf{a}^*||^2 = ||\mathbf{w}^*||^2 = \sigma_a^2 = \sigma_w^2 = 1$ (see Supplementary Material (3). for more generic expressions). The quantity $q_{\text{train}}$ is exactly one in the equivalent kernel limit, and its perturbative correction (in $1/n$) can be found in Supplementary Material (3).

Solving the above equation for $\alpha$, one obtains $l_*$ and hence $\Sigma$ and also $Q_f$ (via Eq. (8)). Using the obtained, $Q_f$ one can calculate the DNN's predictions on the test-set. The effect of the FLS is evident in the second equation, where it controls the deviations from the GP limit. Here we also recall that $\alpha^2$ is the train MSE over $\sigma^4$, thus $\chi_2$ as defined above, contains the MSE factor mentioned in the introduction.

To test the theoretical predictions, we trained two such CNNs, with $n = \{800, 1600\}$; $S = 64$; $N = 20$ and varying channel number. Figure 3, left panel, shows the empirical test-set values for $\alpha$ (dots) compared with a numerical solution of the equations of state (solid lines) and their analytical solution (dashed lines). For the latter, we obtained $\Sigma$ analytically and performed the resulting GP inference with $Q_f$ numerically. The inset tracks the train-set results which, in this case, are fully analytical and involve no numerical GP inference. Both predictions match empirical values quite well, even in the regime where test root MSE is roughly half that of a Gaussian Process ($C \to \infty$). The right panel shows the input layer weights, dotted with $\mathbf{w}^*/|\mathbf{w}^*|$ and with a normalized random vector. These remain Gaussian up to minor statistical noise. Further details can be found in the methods section.

To emphasize the non-perturbative nature of our Eq. (9), let us assume for the sake of negation that they agree with first order perturbation theory in $1/C$ (as in refs. [3,6,23,24]). If so, we may replace $\alpha$ in the above expression for $\chi_2$ by its GP value, as it already contains one negative power of $C$ and hence receives no further corrections at that order. Numerics show this value $\alpha_{GP} = 0.558$ for $n = 1600$. Plugging this in, one obtains $l_* = \frac{2}{S}[1 - 633.2/C]^{-1}$. Clearly, this logic leads to a contradiction unless $C \gg 633.2$. In contrast, our theory provides highly accurate predictions for $n = 1600$, $C = 320$ and $C = 640$ well away from where $\frac{2}{S}[1 - 633.2/C]^{-1}$ admits a perturbation theory in $1/C$. In Supplementary Material (6.2) we report additional results on $l_*$ over its GP value.

**Extensions to Deeper CNNs and Subsets of Real-World Data-sets**

For truly deep CNNs and real-world datasets, obtaining fully analytical predictions for DNN performance is a challenging task, even in the $C \to \infty$ limit. Still, the EoS could be solved numerically and compared with experimental values. Furthermore, the quantities which underlie them could be examined and reasoned upon. We do so here in two richer settings, a 3-layer CNN trained with a teacher CNN and the Myrtle-5 CNN[42] trained on a subset of CIFAR-10. Our first setting extends that of the previous subsection by having an extra activated layer and a target function generated by a similar non-linear teacher network, having however a single channel. For more details, see section "Three layer CNN model definition".

As the first test of our theory, we examine the fluctuations of pre-activations in the input and middle layers of the trained student CNN and check their normality. Specifically, for the input weights $\mathbf{w}_c$ we obtain the histogram (over channels, equilibrium samples, and seeds) of $\mathbf{w}_c \cdot \mathbf{w}^*/|\mathbf{w}^*|$, where $\mathbf{w}^*$ is the teacher input weight and the histogram of $\mathbf{w}_c \cdot \mathbf{w}_r/|\mathbf{w}_r|$ where $\mathbf{w}_r$ is a random vector. Teacher overlap has a variance of 0.254 here, whereas random overlap variance, averaged over choices of $\mathbf{w}_r$'s, was 0.039 with a std of 0.0043. For the hidden layer, we obtain the histogram of $\mathbf{h}_c^{(2)} \cdot \mathbf{h}^{(2),*}/|\mathbf{h}^{(1),*}|$ where $\mathbf{h}^{(2),*}$ are the teacher's pre-activations as well as $\mathbf{h}_c^{(2)} \cdot \mathbf{h}_r^{(2)}/|\mathbf{h}_r^{(2)}|$ where $\mathbf{h}_r^{(2)}$ is the pre-activation of a different randomly chosen teacher. Teacher overlap variance here was, 64.4 whereas average student variance was, 2.3 with a std of 0.12. Figure 4 shows the associated histograms along with their fit to a Gaussian. The large and consistent differences in the variance of the fluctuations between teacher directions and random directions show that we are deep in the feature learning regime. Remarkably, the fluctuations remain almost perfectly Gaussian. The larger variance along teacher directions implies that by drawing DNNs from the

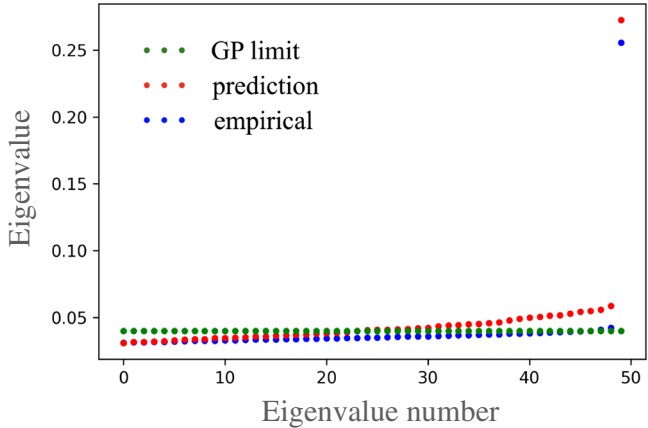

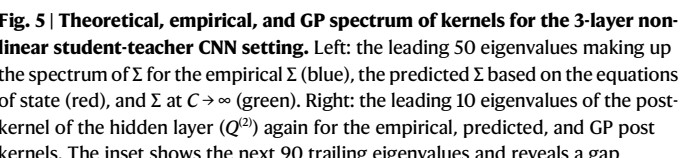

**Fig. 5 | Theoretical, empirical, and GP spectrum of kernels for the 3-layer non-linear student-teacher CNN setting.** Left: the leading 50 eigenvalues making up the spectrum of $\Sigma$ for the empirical $\Sigma$ (blue), the predicted $\Sigma$ based on the equations of state (red), and $\Sigma$ at $C \to \infty$ (green). Right: the leading 10 eigenvalues of the post-kernel of the hidden layer ($Q^{(2)}$) again for the empirical, predicted, and GP post kernels. The inset shows the next 90 trailing eigenvalues and reveals a gap

separating the leading $S_1$ eigenvalues from the rest. The dimension of $Q^{(2)}$ here is $Nn = 2000$, we focus on large eigenvalues as these dominate the predictions. The outlier in the left panel is aligned with the teacher weight vector $\mathbf{w}^*$. The 30 (or $N$) outliers on the right panel are again a feature learning effect and represent the linear feature proportional to $\mathbf{w}^*$ in each of the $N$ latent pixels of the hidden layer.

trained DNN ensemble and diagonalizing their empirical covariance matrices, one is more likely to find dominant eigenvalues along these teacher directions.

We turn to verify the EoS and rationalize the behavior of the pre-kernels. To this end, we average the empirical pre-activations, over channels and training seeds, to obtain an estimator for the pre-kernel and post-kernel of $h^{(2)}$ (i.e. $Q^{(2)}$ and $K^{(2)}$) and that of the input weights ($\Sigma$). We then obtain $\partial D_{KL}(K^{(2)}||Q^{(2)})/\partial\Sigma$ analytically using the 3rd equation from Eqs. (5), plugging in the empirical $\Sigma$. Finally, we compare the empirical $\Sigma$ with that obtained from the last equation from Eqs. (5). Figure 5 left panel plots the eigenvalues of $\Sigma$, our predictions ($\Sigma_{pred}$), and the post-kernel of the input layer which is simply $\sigma_w^{-2} S_0 I_{S_0}$, showing a good match between the first two. Figure 5 right panel plots the eigenvalues of $Q^{(2)}$ as predicted from $K^{(2)}$, compared with its empirical value.

Next, we trained the myrtle-5 CNN[42], capable of good performance and containing both pooling layers and ReLU activations, with $C = 256$ on a subset of CIFAR-10 ($n = 2048$). Barring some one-dimensional integrals requiring numerical evaluation, our EoS generalize straightforwardly to ReLU activation, in which case we are required to track both $K$ (variances) and means (See Supplementary Material 5). Thus, for ReLU feature, learning can manifest itself through both these quantities. Still, the results below suggest that changes to kernels play a dominant role (see also Supplementary Material 6.3). Further analysis of the impact of activation functions on feature learning is left for future work.

Figure 6 shows histograms of various linear combinations of pre-activations. These show a strong deviation of trained DNNs from non-trained DNNs or DNNs at infinite channel/width, and at the same time show quite a good fit to Gaussian in most cases. This opens the possibility of reverse engineering the pre-kernels governing this trained network and using them to rationalize about the DNN, for instance by identifying their dominant eigenvectors.

The 2nd layer (as well as the input layer, (see Supplementary Material (6.3))) show deviations from Gaussianity in the leading eigenvalue. This is expected since the kernels of these layers show quite a dilute dominant spectrum, whereas VGA requires a contribution from many adjacent modes (see Supplementary Material (1.3)). Interestingly, despite this non-Gaussianity in the leading eigenvalue of layers 1 and 2, Gaussainity is restored in the downstream layers 3 and 4.

Correlations across layers and across channels within the same layer are very weak (largely on the order of $10^{-3}$) and fully consistent with the mean-field decoupling underlying this work. Further technical details are found in Supplementary Material (6.3).

## Discussion

In this work, we presented what is, to the best of our knowledge, a novel mean-field framework for analyzing finite deep non-linear neural networks in the feature learning regime. Central to our analysis was a series of mean-field approximations, revealing that pre-activations are weakly correlated between layers and follow a Gaussian distribution within each layer with a pre-kernel $K^{(l)}$. Using the latter together with the post-kernel $Q^{(l)}$ induced by the upstream layer, explicit equations-of-state (EoS) governing the statistics of the hidden layers were given. These enabled us to derive, for the first time, analytical predictions for the performance of non-linear CNNs and deep non-linear FCNs in the feature learning regime. We further note that our EoS generalizes straightforwardly to combined CNN-FCN architectures, ReLU activation functions (see Supplementary Material 5), pooling layers, and models with multiple outputs. The GP represented by the equation we find is a good approximation to the true posterior distribution generated by a large but finite-width Bayesian neural network. Thus, among the other advantages of BNNs, providing reliable uncertainty estimates and principled model comparison[33], they may also admit a concrete interpretation through an effective GP.

Various aspects of this work invite further study. Empirically, it would be interesting to better characterize the scope of models for which a Bayesian sampler (or potentially ensemble-averaged NTK dynamics) leads to Gaussian pre-activations and overall GP-like behavior. Probing the "feature-learning-load" of each layer, by experimentally measuring the differences between the kernels $Q^{(l)}$ and $K^{(l)}$, may also provide insights on generalization, transfer learning, and pruning, thus complementing other diagnostic tools suggested recently[43]. For instance, transferring a layer with a small feature learning load may provide little benefit, and pruning a channel having a large overlap with a leading eigenvalue of $K^{(l)} - Q^{(l)}$ may be harmful.

From the theory side, it is desirable to develop analytical techniques for solving the EoS as well as guarantees regarding the existence and uniqueness of solutions. In particular, exploring the possibility of spontaneous symmetry breaking of internal symmetries such as

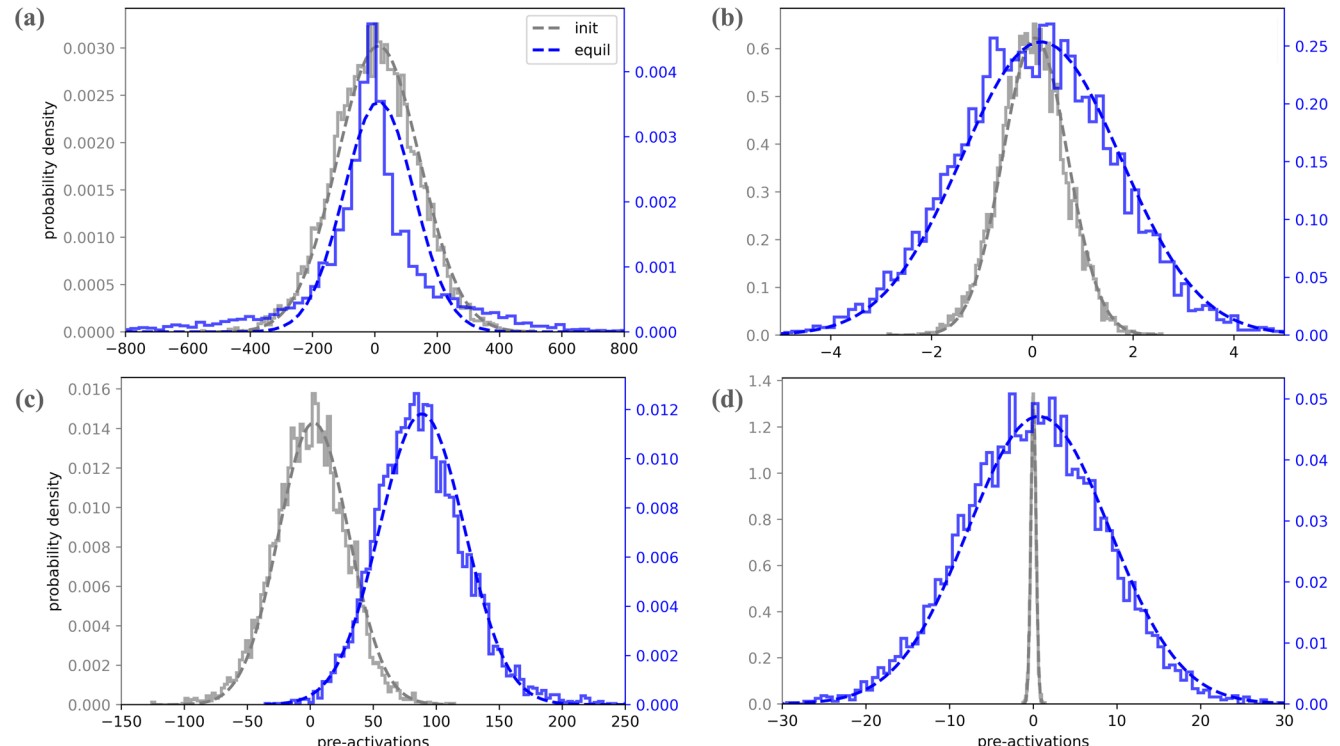

**Fig. 6 | Pre-activation statistics for the myrtle-5 CNN with ReLU activation and $C = 256$, trained on $n = 2048$ CIFAR-10 images. a, b** Statistics of pre-activations of untrained (gray) and trained (blue) Myrtle-5 nets, in the 2nd layer. The histograms are aggregations over channels and initialization seeds of the pre-activations, projected on various eigenvectors of the corresponding kernel matrix. **a** A projection on the 1st (leading) eigenvector, while **b** is the same for the 10th eigenvector. **c, d** Are the same as **a, b**, but for the 4th layer. The fits to a Gaussian (dashed lines) provide an accuracy measure for our variational Gaussian approximation. Notice that the equilibrium distribution, even when it is near Gaussian, can differ

from the initial one in either or both its mean and variance, a signature of feature learning. Generally, we find that Gaussianity increases with the depth of the layer (i.e. downstream layers are more Gaussian than upstream layers), and with the index of the eigenvector, we project on (e.g. here projection on the 10th eigenvector is more Gaussian than on the 1st eigenvector). More details can be found in Supplementary Material (6.3), where we also show that correlations are small across different layers and across channels within each layer, thus justifying neglecting these in our theory.

weight inversion. Providing a mathematical underpinning for the approximations involved here may lend itself to developing performance bounds on the Langevin algorithm and Bayesian neural networks[12,44]. Similarly, one can consider using the empirical effective kernel ($Q_f$) as a starting point to develop GP-based bounds[45] on performance. Last, it is interesting to explore the approach to equilibrium of the training dynamics and adapt the approximations carried here to the NTK setting[1,27].

## Methods

### Mean field action

Here we present the main ingredients of our theory, leading to the EoS we find. Further details can be found in the Supplementary Material.

**Decoupling of layers and neurons.** First, we provide decoupling of Eq. (3) into layer-wise neuron-wise terms, wherein each of the terms depends on the upstream and downstream layers only through channel-averaged second cumulants of activations and pre-activations (pre-kernel and post-kernel). Further details are found in Supplementary Material (1).

Consider the non-linear terms in action in Eq. (3) which couple the different layers. This coupling is mediated through the channel/width-averaged quantities: indeed, $h^{(1)}$ depends on $h^{(2)}$ through the channel/width averaged square term in $h^{(2)}$, $h^{(2)}$ depends on $h^{(1)}$ through the average of $\phi(h^{(1)})\phi(h^{(1)})$, and $h^{(3)}$ depends on $h^{(2)}$ through the average of $\phi(h^{(2)})\phi(h^{(2)})$ and so forth. For, $N_l \gg 1$ we expect these to be weakly fluctuating and well approximated by their mean-field values. This behavior propagates till the output layer, and in particular implies that

the outputs f fluctuate in a Gaussian manner, as previously conjectured[23]. As for the dependency of $h^{(L-1)}$ on the f variables, it is not through a channel/width averaged quantity. However, we find that in various scenarios, such as FCNs with MF scaling or CNNs with large $N$, the fluctuations of f are suppressed enabling us to replace f by its average (see Supplementary Material (1.7) and (2.2)). Following this, we obtain our mean-field action,

$$\mathcal{S}_{\text{MF}} = \sum_{l=1}^{L-1} \mathcal{S}_{\text{MF}}^{(l)} + \mathcal{S}_{f,\text{MF}}. \quad (10)$$

This allows us to define the $\langle \ldots \rangle_{\text{MF}}$ with respect to the distribution: $\pi_{\text{MF}}(\{h^{(l)}\}_{l=1}^{L-1}, f) = e^{-\mathcal{S}_{\text{MF}}}/\mathcal{Z}_{\text{MF}}$, and the partition function $\mathcal{Z}_{\text{MF}} = \int \prod_{l=1}^{L-1} dh^{(l)} df \pi_{\text{MF}}(\{h^{(l)}\}_{l=1}^{L-1}, f)$. Such that

$$\mathcal{S}_{f,\text{MF}} = \frac{1}{2\sigma^2} \sum_{\mu} (f_{\mu} - y_{\mu})^2 + \sum_{\mu\nu} \frac{1}{2} f_{\mu} [Q_f]_{\mu\nu}^{-1} f_{\nu} \quad (11)$$

$$\mathcal{S}_{\text{MF}}^{(l)} = \frac{1}{2} N_{l+1} \sum_{\mu\nu} A_{\mu\nu}^{(l+1)} \bar{Q}_{\mu\nu}^{(l+1)}(h^{(l)}) + \sum_{\mu\nu j} \frac{1}{2} h_{\mu j}^{(l)} [Q^{(l)}]_{\mu\nu}^{-1} h_{\nu j}^{(l)} \quad \text{for } l \in [1, L-1] \quad (12)$$

where $A^{(L)} = \sigma^{-4}(\mathbf{y} - \bar{\mathbf{f}})(\mathbf{y} - \bar{\mathbf{f}})^{\top} - [Q_f + \sigma^2 I_n]^{-1}$, and $A^{(l)} = [Q^{(l)}]^{-1}(I_n - \langle \mathbf{h}_j^{(l)}(\mathbf{h}_j^{(l)})^{\top} \rangle_{\text{MF}}[Q^{(l)}]^{-1})$. The post-kernels are defined self-consistently as $Q_f = \langle \bar{Q}_f \rangle_{\text{MF}}$, and $Q^{(l)} = \langle \bar{Q}^{(l)} \rangle_{\text{MF}}$ for $l \in [1, L-1]$.

Notably, any coupling between the different layers is only through static mean-field quantities, namely the pre-kernels and-post kernels.

In addition, all neuron-neuron couplings (and similarly, channel-channel couplings for CNNs) have been removed.

**Intra-layer decoupling.** Despite the simplified inter-layer coupling and intra-layer neuron coupling, the mean-field actions are still non-quadratic for all layers but the output layer. This non-linearity couples all the $h^{(l)}$ variables for the same neuron (channel in the CNN case) in a way that is roughly all-to-all in the data-point index. In atomic and nuclear physics, similar circumstances are well described by self-consistent Hartree-Fock approximations[46–49]. In our setting, this approximation is directly analogous to a variational Gaussian approximation (VGA). In Supplementary Material (4) we argue that in the typical case where the diagonal of $K^{(l)}$ is much larger than the off-diagonal elements, the VGA is well controlled. Technically, we do so by showing, order by order in perturbation theory, that the diagrams accounted for by the VGA approximation dominate all other perturbation theory diagrams. In Supplementary Material (3) we also establish this using different means for $S_0 \gg 1$ for the specific case of two-layer CNN with a single activated layer. We further comment that the VGA is exact for deep linear DNNs.

Accordingly, we now look for the Gaussian distribution, governed by a kernel $K^{(l)}$ which is the closest to the above non-quadratic action. In models with many hidden layers, this leads to the following "inverse kernel shift" behavior for, $1 < l < L - 2$

$$[[K^{(l-1)}]^{-1}]_{\mu\nu} = [[Q^{(l-1)}]^{-1}]_{\mu\nu} + \frac{2N_l}{N_{l-1}} \frac{\partial D_{KL}(K^{(l)}||Q^{(l)})}{\partial [K^{(l-1)}]_{\mu\nu}} \quad (13)$$

where $l$ denotes a layer index and $D_{KL}(A||B)$ is the Kullback-Leibler (KL) divergence between two multivariate Gaussians with covariance matrices $A$ and $B$. As shown in Supplementary Material (1), for antisymmetric activation functions, the derivative of the KL-divergence is explicitly given by $\text{Tr}[[Q^{(l)}]^{-1}(K^{(l)} - Q^{(l)})[Q^{(l)}]^{-1}(\partial Q^{(l)}/\partial K^{(l-1)}_{i\mu,j\nu})]$. For non-anti-symmetric ones, see Supplementary Material (5).

### Three layer CNN model definition

For the analysis in section "Extensions to deeper CNNs and subsets of real-world data-sets", we consider a student 3-layer CNN defined by

$$f(\mathbf{x}) = \sum_{j=0}^{N-1} \sum_{c'=1}^{C_2} a_{c'j} \phi\left(h_{c'j}^{(2)}(\mathbf{x})\right)$$

$$h_{c'j}^{(2)}(\mathbf{x}) = \sum_{i=0}^{S_1-1} \sum_{c=1}^{C_1} v_{c'ci} \phi\left(h_{cji}^{(1)}(\mathbf{x})\right) \quad (14)$$

$$h_{cji}^{(1)}(\mathbf{x}) = \mathbf{w}_c \cdot \mathbf{x}_{i+jS_1}$$

Where $\mathbf{w}_c, \mathbf{x}_{i+jS_1} \in \mathbb{R}^{S_0}, a \in \mathbb{R}^{C_2 \times N}, v \in \mathbb{R}^{C_2 \times C_1 \times S_1}$, and the input vector $\mathbf{x} \in \mathbb{R}^d$ with $d = NS_1S_0$, and the activation function, $\phi : \mathbb{R} \to \mathbb{R}$, is applied element-wise. See Fig. 7 for illustration. Similarly to before, the regression target $(y_\mu)$ is generated by a random teacher CNN $(y_\mu = f^*(\mathbf{x}_\mu))$ having the same architecture as the student, only with $C_1 = C_2 = 1$. In addition, we chose $S_0 = 50, S_1 = 30, N = 2, C_1 = C_2 = 100$ for the student. We denote teacher weights and pre-activations by * subscripts. Further details are found in the supplementary Information section 2.

### Bayesian posterior sampling with Langevin-type dynamics

In this section, we give more details regarding the algorithm used to generate samples from the Bayesian posterior. We train the DNNs using full-batch gradient descent with weight decay and external white

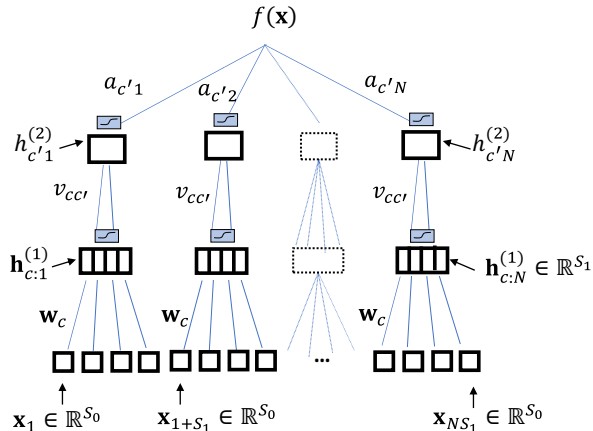

**Fig. 7 | A 3 layers CNN architecture.** An illustration with non-overlapping strides. The black squares represent the strides window in each layer.

Gaussian noise. The discrete-time dynamics of the parameters are thus

$$\boldsymbol{\theta}_{t+1} - \boldsymbol{\theta}_t = -\eta\left(\gamma\boldsymbol{\theta}_t + \nabla_{\boldsymbol{\theta}_t}\mathscr{L}(\boldsymbol{\theta}_t, \mathscr{D}_n)\right) + 2\sigma\sqrt{\eta}\boldsymbol{\xi}_{t+1} \quad (15)$$

where $\boldsymbol{\theta}_t$ is the vector of all network parameters at time step $t$, $\gamma$ is the strength of the weight decay (which, from the Bayesian perspective, is inversely proportional to the variance of the parameters for the prior), $\mathscr{L}(\boldsymbol{\theta}_t, \mathscr{D}_n)$ is the loss as a function of the DNN parameters $\boldsymbol{\theta}_t$, and data, $\sigma$ is the magnitude of noise, $\eta$ is the learning rate and $\xi_{t,i} \sim \mathcal{N}(0,1)$. As $\eta \to 0$ these discrete-time dynamics converge to the continuous-time Langevin equation given by $\dot{\boldsymbol{\theta}}(t) = -\nabla_{\boldsymbol{\theta}}(\frac{\gamma}{2}\|\boldsymbol{\theta}(t)\|^2 + \mathscr{L}(\boldsymbol{\theta}(t), \mathscr{D}_n)) + 2\sigma\boldsymbol{\xi}(t)$ with $\langle \xi_i(t)\xi_j(t') \rangle = \delta_{ij}\delta(t - t')$, such that as $t \to \infty$ the DNN parameters $\boldsymbol{\theta}$ will be sampled from the equilibrium Gibbs distribution in parameter space $p(\boldsymbol{\theta}|\mathscr{D}_n)$. In principle, this is true regardless of the initial condition given to the parameters. However, in practice, to achieve reasonable convergence times we set a random initial condition with zero mean and a variance that matches that of the Bayesian prior.

This algorithm, used to generate samples from the Bayes posterior, corresponds to the Unadjusted Langevin Algorithm (ULA)[50–52], together with a weight decay term. Hyperparameters such as learning rate, weight decay, and noise level (or mini-batch size for SGD) can be experimented with and compared across different training protocols, thus making ours more tightly connected to gradient-based algorithms used by practitioners for training DNNs, while also conforming to the Bayesian perspective. While this method of posterior sampling may be slower to converge compared to other more sophisticated samplers (such as HMC[33]), it is simpler (e.g. has no Metropolis acceptance step, hence the word "unadjusted") and admits an intuitive correspondence with vanilla SGD, where the mini-batch noise is replaced with white additive noise[31]. Under some rather mild conditions, ULA has been shown to have good convergence properties[50].

### Experimental details

**Hyperparameters.** For the 2-layer CNN experiments, we used $S = 64$, $N = 20$, and varying channel number. The training parameters (noise and weight-decay) were tuned such that $\sigma^2 = 0.1$ and weight variance of 2.0 over fan-in, for both layers at $n = 0$. The target was drawn once for all experiments using i.i.d. Gaussian centered random $a_i^*$ and $w_s^*$ with variances $1/N$ and $1/S$ respectively.

For the 3-layer CNN experiments, we took $S_1 = 50, S_0 = 30, N = 2$. The training parameters (noise and weight-decay) were scaled such that $\sigma^2 = 0.005$ and weight variance of 2.0 over fan-in for the inputs and

hidden layer with no training data (at initialization). The weight variance of the read-out layer was 15 over the fan-in. The target was drawn again once for all experiments from a teacher CNN with $C = 1$.

For all the myrtle-5 experiments, we used $n = 2048$, $C = 256$ and ReLU activation. The training parameters (noise and weight-decay) were scaled such that $\sigma^2 = 0.005$ and weight variance of 2.0 over fan-in for all layers with no training data (at initialization).

For all the FCN experiments, we used equal width ($N_1 = N_2$) and weight decay corresponding to variance, $\sigma_w^2 = \sigma_a^2 = 2$ (with no training data) in the regular scaling. For the MF scaling, we took $\sigma_a^2 = 2/N_2$. The target was drawn again once for all experiments from a teacher CNN with $N_1 = N_2 = 1$. Specifically, when calculating the emergent scale, we used $\sigma_a^2 = 2/256$ independent of $N_2$.

**Equilibrium sampling.** To obtain weakly correlated samples from the equilibrium distribution of the trained CNNs we used the following procedure. For the 2 and 3-layer CNNs, we used an adaptive learning rate scheduler: For the first 100 epochs we used a learning rate $lr_0/10$, then we crank up the learning rate to $lr_0$. As of epoch $5e3$, every $1e3$ epoch we estimate the fluctuations of the train-loss and check for spikes—events in which the train-loss was five times larger than the standard deviation in the past 500 epochs. If a spike is observed, the learning rate is reduced by a factor of 0.7. This continues until $5e4$ epochs pass without any events. Then the learning rate is reduced again by a factor of two and remains fixed. Samples from these final stages were treated as equilibrium samples. We further checked that (i) different initialization seeds trained with this protocol reached the same train-loss statistics. (ii) No further reduction in train-loss occurred after the final learning rate reduction. For several runs, we also verified that increasing the last reduction of learning rate by an additional factor of 2 did not have any appreciable effect on the loss. The initial $lr_0$ was $\sim 1e-4$ (w.r.t. a standard mean reduction MSE loss) and the final learning rate was typically $\sim 1e-5$. The runs terminated at epoch $3e5$.

For the myrtle-5 CNN trained on CIFAR-10, we first ran several runs for $3e5$ epochs using the above procedure and examined those that reached the lowest train-loss. We then generated a fixed scheduler based on those more successful instances, running up to $4e5$ epochs. We again verified that further lowering the final learning rate has no appreciable effect on the training loss, and that different seeds reach similar final train-loss. This ensures that we are indeed sampling from a valid equilibrium distribution.

For the 3-layer CNN and Myrtle-5, we found that auto-correlation times of pre-activations change considerably between the layers. While the read-out layer typically had an auto-correlation time of the order of $1e3$ epochs (at the lowest learning rates) the auto-correlation times for the input layers could reach $\sim 1e6$ or larger values. To overcome this issue, when analyzing pre-activations of these deeper DNNs we took an ensemble containing 98 and 234 different initialization seeds for the 3-layer CNN and Myrtle-5 respectively.

For the 3-layer FCN we used a fixed scheduler which starts at 1/2 the maximal stable learning rate and reduces the learning rate by factors of 2 at 100, $1e5$, $1e6$, $3e6$ epochs and by a factor of 4 at $4e6$, $5e6$ epochs (factor of 128 in total). Equilibrium sampling was done between $6e6 - 7e6$ epochs.

**Numerical solution of the equations of state.** For the 2-layer CNN, the equations of state were solved using Newton-Krylov method[53], which does not require explicit gradients. To facilitate convergence, we adopted an annealing procedure: For $C \sim 1e3$, we obtain the solution using a GP initial value ($x_0$) for $\Sigma$. The optimization outcome was then used as $x_0$ for the next lower value of $C$. Using 12 CPU cores, this optimization took several hours. After obtaining $\Sigma_{ss'}$ as a function of $C$, the resulting kernel $[Q_f]_{\mu\nu}$ was used in standard GP inference to obtain $f$ on the test-set. For the 3-layer FCN we used a more efficient JAX-based

code to generate the kernels and kernel derivatives involved in the EoS, but otherwise followed the same procedure. Optimization took between several minutes to a few hours on one Titan-X GPU, depending on parameters.

## Data availability
The datasets used in this study are either publicly available online (such as the CIFAR dataset) or can be generated by the code found in the following repository https://github.com/zringel/AdaptiveGPs.

## Code availability
The code used to perform numerical experiments, analyze data, give theoretical predictions and generate figures is available in the following repository https://github.com/zringel/AdaptiveGPs.

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

## Acknowledgements

We acknowledge support from Gatsby Charitable Foundation (G.N.), the Swartz Foundation (G.N.), the National Institutes of Health grant No. 1U19NS104653 (G.N.), and the MAFAT Center for Deep Learning (G.N.). I.S. was partially supported by the Israel Science Foundation grant 421/20.

## Author contributions

The theoretical model was developed by I.S. and Z.R. with inputs from G.N. Numerical simulations and visualization of results were carried out by Z.R. and G.N. The manuscript was written and edited by all authors.

## Competing interests

The authors declare no competing interests.
