## [Peer Review File · Nature Communications]

Reviewer comments, first round -

Reviewer #1 (Remarks to the Author):

This paper is describing a new mean field theory of deep convolutional neural networks that claims to be valid beyond the infinite-width limit of neural networks. If true, this would be an important advance because one could address feature learning, which does not happen in the infinite-width limit. However, the current submission is not ready to be published, and not sufficient for a publication in a journal with a broad audience such as Nature Communications. I list my reasons below.

1. First, the current manuscript makes heavy use of physics jargon, and is not readable for either a machine learning audience or the broader audience of Nature Communications. Just as an example, let's take equation 2, to which authors lead by the phrase "As shown in Supp. Mat. (1) training such a CNN using the above training procedure leads to a partition function governed by the following action:". Now, there are many things going on in this sentence that wouldn't be obvious to only specialists. Authors are making a connection between Bayesian posterior, a concept from machine learning and the object that they are studying, to partition functions and actions, concepts from statistical physics. However, the words Bayesian posterior never occur in the main text and the connection is never made explicit.

I would claim that even theoretical physicists would find this paper hard to read. Beyond removal of jargon, clarity of writing needs to be significantly improved (as well as paying attention to typos and missing references).

2. Related to the point above, the framing of the manuscript is puzzling and somewhat misleading. It should focus on the Bayesian setting rather than presenting it as "GD+Noise," since the theory is purely concerned with the equilibrium distribution.

3. The paper focuses more on the methodology of building the mean field theory, rather than describing the insight gained from it. This is a major weakness. Further, sections 2.1 and 2.3 are not describing results, but assumptions (see below) and methods.

4. The paper derives the mean field theory in a fairly obscure way, which prevents me from giving an expert opinion on its validity.

a) The hierarchy of approximations is not presented clearly. This is the most frustrating part in reading the SI.

b) The MF average is not explicitly defined as far as I can see. This is critical because the "decoupling" result the authors are presenting is merely ignoring fluctuations in the kernels and replacing them by their "MF average".

c) It should be made clear that Gaussianity of the kernels relies on large width. Moreover, the authors do not quantitatively test for Gaussianity. They should show higher cumulants, ideally as a function of width and sample size.

d) The importance of odd activation function ($\phi(-x) = -\phi(x)$) should be made clear.

e) As far as I can see, there is no reason that this theory would not apply to multilayer perceptrons. That version could be easier to present.

5. The paper should discuss how these approximations relate to other works addressing feature learning beyond the infinite width limit, e.g. variational approximations for BNN posteriors, c.f. <https://arxiv.org/abs/2202.11670>, and various papers attempting perturbative expansions around the infinite-width limit.

6. Simulation results are not sufficient. The theory makes detailed predictions about the nature of learned representations, however theoretical kernels are not compared to simulations. How do these results compare to perturbation theory in $1/\text{width}$? It is not clear what the point of Figure 5 is.

7. The resulting mean field theory requires substantial numerical procedures. The computational complexity of solving this mean field theory should be compared to training the neural network.

Reviewer #2 (Remarks to the Author):

This paper addresses the question of how representations are learned in CNNs. To my mind, this is perhaps the key theoretical problem in understanding CNNs, so it is exciting to see the authors' work in this area. This paper takes a statistical physics approach, and constitutes a big advance in our understanding of these questions over past work (e.g. Zavatone-Veth et al 2021). My comments are relatively minor: concerning the presentation of equations in the main text and relationship to a broader body of work (that does not bring novelty into question).

While it is absolutely the author's choice, I prefer the "Gram matrix" for the pre-activation kernel, and simply "kernel" for the post-activation kernel (see [1]). The logic behind this choice is that we can treat the network as a BNN, and a BNN is equivalent to a Gaussian process, with a kernel given by the product of the post-activations with themselves (the author's free kernel). At the very least, it would be good to have a bit more justification on the choice of these terms. I have to look up which one we're referring to every single time. Even pre-kernel and post-kernel would be easier to remember.

"Still, the task of finding a useful set of variables that track this flexibility and obey an explicit, general, and closed set of equations — remained unresolved."

There are a couple of nuances here. First, it has been evident for a while through a very large body of work that the right way of describing representations and flexible representation learning in NNs and other deep nonlinear function approximators is going to be through the kernels. For instance, [1] shows that the BNN generative model can be written entirely in terms of just the (free) kernels, without needing to represent the latent kernels, features or weights. This implies that the (free) kernels are the only necessary set of variables to track flexible representation learning. (Note that [1] differs in that it doesn't give the closed-form set of equations). Second, the [2] paper gives a GP limit with explicit, general and closed form set of equations for understanding representation learning in deep flexible non-linear function approximators. This is not the same result as the present paper, because [2] is restricted to deep GPs, which are closely related to NNs, but are not equivalent. That said, [2] was out on arXiv only a couple of months before this article, so it might be okay to ignore it: maybe one for the editor. [2] also has consequences for statements like "GP limits miss out on several qualitative aspects, such as feature learning". Overall, I'd suggest a formulation something like: "Still, the task of providing an explicit, general closed set of equations describing this flexibility in NNs remains unsolved."

"This stands in contrast to the view that data-points, in latent space, cluster according to labels (compression of relevant information)."

This is very strange. Representation learning in neural networks are known to do precisely this: cluster data-points according to labels. Indeed, empirically neural networks usually end up eliminating all information except that relevant to the label. So for instance for CIFAR-10, you end up with a 10-dimensional top-layer representation. This can be seen empirically, and with various elegant descriptions in linear networks [3]. I think the error is here: "The above also implies that any pre-activation is as likely as any of its rotations in channel space." There is indeed an elegant rotational symmetry (e.g. [1]), but you have to rotate the whole pre-activation space (i.e. all datapoints simultaneously). Critically, rotating all datapoints retains the clustering structure.

Introduction iii: n is not yet defined? And (assuming that n is the number of datapoints), the number of datapoints (squared!) is typically much larger than the number of channels, so wouldn't the emergent scale typically be larger than $O(1)$?

In the main text, results are given for convolutional networks. However,

1. As far as I can see, the theory presented is still novel for fully connected networks (i.e. the authors don't just present an extension of existing fully-connected theory to CNNs).
2. Results on fully connected networks are much simpler, while still capturing all the key ideas. I'd therefore suggest changing all(?) equations in the main text to fully connected, and leaving the

CNN extension to Methods/Supplementary (but obviously keeping the CNN experimental results in the main text).

Eq. 1

The terms in this equation need to be considerably better described:

1. Can we say explicitly that a , v and w are weights?
2. Can we say explicitly that i, j index spatial location?, and what (spatial?) dimensions are of size S_0 , S_1 and N .
3. Notation for x is a mess: \mathbf{x}_μ is a datapoint. And at the same time, \mathbf{x}_{i+jS_1} is ... something else? Maybe a patch?
4. In the main text, can we drop σ_a , σ_v , by taking the weights to be IID Gaussian with variance $1/\text{fan-in}$? Fewer symbols is preferable. At the very least, the a in σ_a looks like an index, and should be σ_{a} if it is a label.

Eq. 4

"The above also suggests that the low-lying spectrum of $Q(l)$ remains relatively inert even in presence of the back-propagation term and tends to align with that of $K(l)$." isn't very obvious to me from the presented equation. Is there any way to make this more evident?

"Extensions to deeper CNNs and real-world datasets"

Slightly misleading given that we're using a very small subset of CIFAR-10. I'd go with something like "Extensions to deeper CNNs and subsets of real-world datasets".

The analysis of the similarity of the kernel matrices is a bit limited. e.g. is it possible to show convergence using e.g. the Frobenius norm as the width increases?

"While the read-out layer typically had an auto-correlation time of the order of 10^3 epochs (at the lowest learning rates) the auto-correlation times for the input layers could be $\sim 10^6$ or larger. To overcome this issue, when analyzing pre-activations of these deeper DNNs we took an ensemble containing 98 and 234 different initialization seeds for the 3 layer CNN and Myrtle-5 respectively."

Ensembling is not a valid solution to slow convergence. The only thing to do is to run longer or find some other trick to improve convergence (e.g. momentum, tempering the noise?). Maybe that's okay given that it's likely to only push you away from theoretical matching. In general, the procedure for sampling is extremely elaborate, which surprises me. Generally, e.g. SGLD (which is similar, but not the same) is usually very well-behaved. Could you comment on the difficulties?

Minor points:

Quite a few broken refs.

" σ is only through the mean and variance of $h(1)$ "

I couldn't find details of the Newton-Krylov method

Capitalization in the refs "27. Seeger, M. Pac-bayesian"

mutatis mutandis (I am not a fan of latin phrases).

[1] Aitchison L, Yang A, Ober SW. Deep kernel processes. ICML (2021)

[2] Yang A, Robeyns M, Schoots N, and Aitchison L. Deep kernel machines: exact inference with representation learning in infinite Bayesian neural networks (arXiv:2108.13097 2021)

[3] Aitchison L. Why bigger is not always better: on finite and infinite neural networks. ICML (2020)

Response to Reviewers Comments

We would like to thank the reviewers for their constructive comments, which helped us improve our manuscript significantly. We have revised the manuscript to incorporate all the comments received and now provide a substantially revised version.

The main changes and additions are the following

1. Based on a suggestion from both the reviewers, we have now successfully applied our theory to fully connected DNNs (FCNs). While the main goal of the paper is still to tackle outstanding issues with CNNs, it turns out that FCNs in the 'Mean-field' (MF) scaling [8] portrays many of the key theoretical aspects in a clean and controlled manner.
2. Following the previous point, and to make the paper more accessible to non-specialists, the revised main text explains our theory for the FCN case and delegates much of the formulae for CNNs (which have more involved index notation) to the appendix. In addition, we separated the methods section from the results section. The methods section now provides an overview of key technical aspects of the work with further details appearing in the appendix. The results section has been made more general and highlights the key conceptual points.
3. The revised appendix contains two new analytical sub-sections (1.6 and 2.2) which study the validity of our approach, by calculating perturbative corrections. These provide extra assurance that our approximation is controlled at a large width/channel number for both FCNs with MF scaling and CNNs.
4. The revised manuscript also fleshes out more clearly the role of the emergent feature learning scale (χ) in controlling the amount of feature learning (in the sense of changes to kernels). In particular, we provide expressions for it for FCNs and CNNs, and evaluate it for all the toy settings we considered (see Fig. 1.). This serves two purposes. First, despite the fact that χ contains a negative power of the width (or channel number) it is very large for CNNs and FCNs with MF scaling (yet small for FCNs in the standard scaling). Its scale thus explains how perturbation theory in $1/\text{width}$ fails: Indeed a perturbation in $1/\text{width}$ entails a perturbation in χ however the latter is often large in the feature learning regime. Second,

it provides a simple measure that does not require solving any complicated equations, for which feature learning effects become dominant in parameter space.

In what follows, we explain how each and every comment was addressed, with references to parts of the paper that have been modified pursuant to the reviewer’s suggestions. Our responses appear in the non-italic font.

Reviewer #1 :

This paper is describing a new mean field theory of deep convolutional neural networks that claims to be valid beyond the infinite-width limit of neural networks. If true, this would be an important advance because one could address feature learning, which does not happen in the infinite-width limit. However, the current submission is not ready to be published and is not sufficient for publication in a journal with a broad audience such as Nature Communications. I list my reasons below.

1. *First, the current manuscript makes heavy use of physics jargon, and is not readable for either a machine learning audience or the broader audience of Nature Communications. Just as an example, let’s take equation 2, to which authors lead by the phrase “As shown in Supp. Mat. (1) training such a CNN using the above training procedure leads to a partition function governed by the following action:” . Now, there are many things going on in this sentence that wouldn’t be obvious to only specialists. Authors are making a connection between Bayesian posterior, a concept from machine learning and the object that they are studying, to partition functions and actions, concepts from statistical physics. However, the words Bayesian posterior never occur in the main text and the connection is never made explicit. I would claim that even theoretical physicists would find this paper hard to read. Beyond the removal of jargon, clarity of writing needs to be significantly improved (as well as paying attention to typos and missing references).*

Response: We put a lot of effort into rewriting and reorganizing our manuscript such that it will be accessible to a wider audience and corrected all missing references and typos we detected. In particular, we unpacked the above statement involving actions, equilibrium, and partition functions. This hopefully explains better the connection between Bayesian posterior and its representation in statistical physics in terms of action and partition function. Our original point of view is to look at the equilibrium distribution of the Langevin (Gradient Descent + noise) algorithm with weight decay, since using this view we can compare our result to experiments. This is also the reason why we did not introduce the Bayesian posterior point of view. The referee is right that these points of view are equivalent in terms of the mathematical object we study. We added further explanation about these two interpretations of our results

in Sec. 1.1. and in the discussion. In addition, we clarify and added an explanation of the statistical physics jargon along the paper.

2. *Related to the point above, the framing of the manuscript is puzzling and somewhat misleading. It should focus on the Bayesian setting rather than presenting it as “GD+Noise,” since the theory is purely concerned with the equilibrium distribution.*

Response: We present in the paper the two points of view, putting now more emphasis on the Bayesian picture. We kept the GD+Noise picture since we can compare it to our simulation scheme and since we believe it would help communicate the results to practitioners better.

3. *The paper focuses more on the methodology of building the mean-field theory, rather than describing the insight gained from it. This is a major weakness. Further, sections 2.1 and 2.3 are not describing results, but assumptions (see below) and methods.*

Response: We thank the referee for this comment. Indeed, the previous manuscript provided a somewhat physics-style presentation where the methodology is given before the results. We have now fully adopted Nat. Comm. style where results are put first. This should also help us address a broader audience as one now does not need to understand partition functions, perturbation theory, or mean-field theory, to appreciate our results.

More specifically, we rearranged the paper, placing our equations of state, defining the data-aware GP process we find, as our main result, and delegating the mean-field decoupling + Variational Gaussian Approximation (VGA) to the methodology section. In addition, in our problem statement section, we now provide intuition for what type of finite-width effects are captured by our treatment. Furthermore, we provide a more detailed discussion about the main insights one gains from these equations. This covers various key points which were previously scattered around the text and the appendix. Specifically, we now discuss

- The non-trivial fact that our EoS provide us with a concrete effective GP description for the entire DNN as well as its hidden layers. A priori one would expect that the normality of pre-activations, a large C, N trait, will be lost at finite C, N . Yet we find that pre-activations remain Gaussian and accommodate strong feature learning effects while maintaining accurate predictions. This unexpectedly simple behavior opens various reverse engineering possibilities wherein one infers the effective kernels from experiments and uses their spectrum and eigenvalues to rationalize about the DNN (see also Fig. 5 in the main text).
- The emergent scale as it arises from the EoS. In particular, our results show that there is a crossover to a feature learning regime which is identified by the emergent scale we find being $O(1)$

- The failure of perturbation theory when the emergent scale is non-negligible
- The output covariance matrix (last layer post-kernel) leading eigenvector’s direction tends to align with the discrepancy of the output
- The contrast between our viewpoint and the compression-of-relevant-information viewpoint [12].

Last but not least, the referee’s suggestion that we look into FCN has helped us a lot since it now allows a single experiment that showcases many aspects of the theory. This is reported in the text (Fig. 1.) immediately after the above discussion.

4. *The paper derives the mean field theory in a fairly obscure way, which prevents me from giving an expert opinion on its validity.*

- (a) *The hierarchy of approximations is not presented clearly. This is the most frustrating part in reading the SI.*

Response: We listed at the beginning of the Results section the approximations we do. We also improved the methods section such that it now provides a more comprehensive overview of the approximations we carry. The Supp. Mat. has also been substantially revised. First by focusing on the simpler FCN case and second by clearing notation.

Turning to the technical rather than presentation issue we note that our approach involves two main approximations: **(a)** Our first approximation is the mean-field decoupling, in which we suppress kernel fluctuations of order $1/(\text{width or channel})$ of each latent layer. We discuss the validity of the mean-field in Supp. Mat. (1.6) for FCN and Supp. Mat. (2.2) for CNNs where we perform perturbation theory and calculate the first order correction (higher moments). The above approximation yields the mean field distribution. This is general and does not require the antisymmetric assumption on the activation function, but does require large width see also Supp. Mat. (1). **(b)** The mean field probability distribution is still not Gaussian, and therefore we look for the closest Gaussian distribution by minimizing the Kullback-Leibler divergence between the mean field distribution and a multivariate Gaussian distribution. We call this the variational Gaussian Approximation (VGA). Here, for simplicity, we consider an antisymmetric activation function which allowed us by symmetry to take a centered distribution for each layer. We discuss the general case in Supp. Mat. (5). We also estimate the validity of the VGA via perturbation theory in Supp. Mat. (4). There we show that the VGA is more accurate than doing perturbation theory in $1/\text{width}$.

- (b) *The MF average is not explicitly defined as far as I can see. This is critical because the “decoupling” result the authors are presenting*

is merely ignoring fluctuations in the kernels and replacing them by their “MF average”.

Response: We now define the mean field average in the methods where we explain the mean field decoupling. (section 4.1.1) in the main text.

- (c) *It should be made clear that Gaussianity of the kernels relies on large width. Moreover, the authors do not quantitatively test for Gaussian. They should show higher cumulants, ideally as a function of width and sample size.*

Response: We provided additional sections (Supp. Mat. (1.6) for FCN and Supp. Mat. (2.2) for CNNs) which explicitly calculates perturbative corrections (higher cumulants) to our mean-field decoupling, thus providing extra confidence that what we deemed negligible based on standard mean-field reasoning, is indeed negligible. The main text (Fig. 1) and Supp. Mat. (6) now show qq-plots as a more precise measure of normality.

- (d) *The importance of odd activation function ($\phi(-x) = -\phi(x)$) should be made clear.*

Response: We added a remark about that in the results section, as well as in several places in the Supp. Mat. including section (5) in Supp. Mat., which discusses how to generalize our theory to a non-anti-symmetric activation function.

- (e) *As far as I can see, there is no reason that this theory would not apply to multilayer perceptrons. That version could be easier to present.*

Response: We now present our equations of state for deep FCN with any finite number of layers (Eq. (4) in the main text). We have added a detailed section (Supp. Mat. (1)) in which we provide a detailed derivation of our theory for deep FCNs. Apart from the simplifications arising from looking at FCN rather than CNNs, we found further ways to clarify and simplify the derivation.

5. *The paper should discuss how these approximations relate to other works addressing feature learning beyond the infinite width limit, e.g. variational approximations for BNN posteriors, c.f. [4], and various papers attempting perturbative expansions around the infinite-width limit.*

Response: Our result provides a closed set of equations (the EoS) that predict the output of the network at the feature learning regime, whereas perturbative results such as [13, 9] have poor convergence in the sense that width/channels have to scale with n . For NTK see [7] figure S5, for BNNs see [9, 10]). This was mentioned in a few places in the older version of our manuscript, but now we are more explicit about this, in particular when we discuss the non-perturbative nature of our results both in the general context of the emergent feature learning scale we find and in more detail where we discuss the 2-layer CNN.

Some other works [5, 6, 11] provide an infinite set of equations that are generally not tractable (indeed many of the outstanding problems in physics (e.g. high-Tc superconductivity, confinement in QCD) would be considered solved if one accepts an infinite perturbative expansion as an answer).

Several works also study the mean-field scaling [14]. This can also be addressed using our framework by scaling the variance of the weights accordingly. Specifically, this is now addressed in our newly reported FCN experiment and theory.

We thank the referee for pointing Ref. [4]. This work and ours are similar in the sense we both, in effect, study a variational approximation to the BNN posterior distribution. Crucially, however, the assumptions on the variational posterior distribution we used are very different. Ref. [4] uses an independent Gaussian distribution for the posterior weights whereas we used Gaussian distribution for the posterior pre-activations which allows intra-layer correlation. Specifically for the input layer, pre-activations and weights are linearly dependent. There we find it more compact to work with correlated multi-variate Gaussian distribution for the weights whose covariance is Σ . Here we find that feature learning effects manifest themselves through correlations between the weights. Specifically, in the example of the 2-layer teacher-student model, the posterior covariance matrix Σ is given by the uncorrelated prior plus a rank one perturbation in the direction of the teacher’s true weights.

Due to this important difference in our approaches, our results differ too. Ref. [4] finds the disturbing result that for odd-activations and infinite width the posterior is the prior. We instead obtain the GP posterior in the same limit which faithfully captures the BNN’s behavior. In addition, our variational approach yields accurate predictions within the feature learning regime. The latter was mentioned as an important need in Ref. [4]. Indeed citing the discussion there: “...*In cases when inaccurate inference is combined with very large models, our results prove a crippling and previously unknown limitation to this approach. This highlights the need for accurate inference methods in Bayesian neural networks ...*”. We now cite this work in the introduction (as well as an earlier work).

6. *Simulation results are not sufficient. The theory makes detailed predictions about the nature of learned representations, however theoretical kernels are not compared to simulations.*

Response: The new manuscript now includes numerical results on 3-layer non-linear Fully Connected Networks (FCNs) within the mean-field scaling (i.e. low variance for the top layer), numerical estimates of the emergent scale for FCNs and CNNs, and results for Myrtle-5 with $n = 2048$ data-points instead of $n = 256$ data points. Last, we agree that for some reason, numerical results about changes to the kernel were somewhat hidden in the previous version. They are now discussed more explicitly. Specifically,

- The new FCN results (Fig. 1.) report the ratio of Σ eigenvalues to those of the GP showing changes by a factor of 2. Supp. Mat. (6) also shows a comparison of the full spectrum of the trained network compared to the GP.
- For the 2-layer CNN experiments, we now report also the ratio of the top Σ eigenvalue to that of the GP. (See Supp. Mat. (6.2)). In addition, Fig. 2 (Fig. 1 in the previous version) compares our analytical prediction for the 2-layer CNN of the discrepancy from the target with the experiment and the GP limit.
- In the 3-layer CNN experiment, we stress that the variance of the histograms, along a direction \mathbf{v} is $\mathbf{v}^T K \mathbf{v}$ and hence reflects changes to the kernel. As the variance of \mathbf{v} along a randomly chosen direction is much smaller than along the teacher’s weights – our results show a sharp signature of strong feature learning effects.
- The point of Fig. 5 (now Fig. 6) is to show whether our theory holds for real-world data-sets and more complex networks such as Myrtle-5. We find that the distributions of pre-activations before and (more importantly and perhaps surprisingly) after training are well described by a Gaussian, at least for the deeper layers. We also found that our mean-field decoupling indeed holds over channels and layers, as we detail in the Supp. Mat (6.3).
- Last, we comment that one place where a direct reference to kernel spectra appeared was Fig. 5. (Fig. 4. of the previous version). The point of this figure was to show that trained 3-layer CNNs follow our equations of state. Specifically, we calculated Σ based on the (CNN version) of the line before last in Eq. (4) in the main text, where we used the empirically sampled $K^{(2)}$ and $Q^{(2)}$. We then compared this with the empirical Σ and the Σ obtained from the GP limit.

7. *How do these results compare to perturbation theory in 1/width?*

Response: This is a very central point, so please allow us to elaborate. As we further clarify now, both in our new results paragraph on the emergent scale and specifically in the 2-layer CNN model – our results are non-perturbative in the emergent feature learning scale which contains a negative power of N_2 or C_2 (the width or channel number of the 2nd layer for FCNs or CNNs respectively). Thus, they are non-perturbative in $1/N_2$ or $1/C_2$. More quantitatively,

- The experimental and theoretical predictions reported in Fig. 2. left panel are normalized by their infinite-channel/GP value. Still, they are around 0.5 or 2 for the train discrepancy and leading Σ eigenvalue respectively. It is unlikely that perturbation theory will capture such large corrections accurately.

- The emergent scale is the second term in the denominator in the equation for l_* [Eq. (8)]. Let us assume, for the sake of negation, that our results are on par with first order perturbation theory in $1/C$. If so, we may replace α (essentially the training error) in that term by its GP value, as this emergent-scale-term already contains one negative power of C . Numerics show this value is $\alpha_{GP} = 0.558$ for $n = 1600$. Plugging this value, along with the values for $\sigma_a^2, \sigma_w^2, \|\mathbf{a}^*\|^2, \|\mathbf{w}^*\|^2, n$ used in this experiment, one finds it is $[4(0.558 \cdot 1600)^2 \cdot 2 \cdot 2 / (C \cdot 64 \cdot 20 \cdot 5 \cdot 3.147)] \cdot 1 \cdot 1.1 = 633.2/C$. Thus, under the assumption that leading order perturbation theory holds, one finds that $l_* = \frac{2}{5}[1 - 633.2/C]^{-1}$. Clearly, this is inconsistent with first order perturbation theory (or low order perturbation theory) unless $C \gg 633$. In contrast, our theory provides highly accurate predictions for $n = 1600, C = 320$ and $C = 640$ well away from where $\frac{2}{5}[1 - 633.2/C]^{-1}$ admits a perturbation theory in $1/C$. We comment that using α found by the solver at $C = 320$ which is $\alpha = 0.309$, the term in the denominator becomes $209.99/320 = 0.656$ leading to a roughly 290% increase in l_* , the leading Σ eigenvalue, compared to its GP value ($\frac{2}{5}$). Similar demonstrations of strong shifts to kernel eigenvalue from their GP limit appear throughout the text. This discussion is now part of the main text.
- The quantity we used in the previous item (specifically $633.2/C$) is what we more broadly refer to now as the emergent feature learning scale. Figure 1 now reports how this scale behaves for various values of n for FCNs with standard scaling, FCNs with MF scaling [8] (i.e. when the variance of the weights is of order $1/N_l$), and CNNs with standard scaling. In both latter cases, it indicates strongly non-perturbative behavior (in the sense of it being $O(1)$ or larger) for channel/width values where our theoretical predictions are highly accurate. For FCNs with MF scaling, at $n = 1024$ the emergent scale is 1 at $N_2 = 10^5$ and thus much larger than 1 at $N_1 = N_2 = 1024$ where we report our analytical predictions. Thus, we are making accurate predictions in the region where perturbation in $1/N_l$ is a perturbation theory in a quantity that is much larger than 1 and hence invalid. In contrast, the results reported in Figure 1. panel (b) show nearly exact results for RMSE and α along with a minor 3.5% error in predicting the top Σ eigenvalue.

Given the above three points, perturbation theory being accurate in the parameter regime we report will be highly unlikely.

- We also point out that prior works by some of the authors have studied the validity of perturbation theory and reported similar results [9] [10]. Specifically, we draw the referee’s attention to Ref. [10] where we studied a two-layer linear CNN and compared the experimental results with perturbation theory (Fig. 2. panel (c)). Our

findings there, clearly support the fact that perturbation theory provides a very poor account for feature learning. In addition, Fig. S5 in Ref. [7] shows similar evidence for the NTK case: as n increases, the width needs to grow as n in order to be in a near-GP regime, where perturbation theory is justified.

8. *The resulting mean field theory requires substantial numerical procedures. The computational complexity of solving this mean field theory should be compared to training the neural network.*

Response: Our equations of state (EoS) reduce the problem of estimating averages under a complex non-linear/non-Gaussian partition function, to the task of solving a set of non-linear equations. In simple cases, these equations lend themselves to a fully analytical solution (the 2-layer cases). We comment that in a future publication with Dr. Claude Fleming, we will substantially expand this set of toy models with analytically tractable EoS.

More broadly, the goal of these equations was not to provide a novel inference algorithm but to provide a clearer analytical picture of feature learning. A similar criticism can be raised regarding the Gaussian Process viewpoint at $N_l \rightarrow \infty$: often storing the matrices involved and inverting them is much harder than training a DNN.

That being said, for FCNs the equations can be solved rather efficiently. As we now report in the text, for the parameters we used, it typically takes between a few minutes to an hour to solve these on one Titan-X GPU. In contrast, training the DNN for many epochs at vanishing learning rates to gain proper equilibrium statistics, takes several hours. For deep CNNs, solving the EoS is harder at the moment, since the pre-kernels (which are the variables we are estimating) have a dimension of n times the number of latent CNN pixels. This quickly leads to matrices whose dimension is of the order of millions. Complexity-wise, we believe that storing and inverting such matrices is the bottleneck, time-wise and space-wise. Still, just like in GP inference, there is much room for improving this. One direction to consider here is to focus on the dominant eigenspace of these pre-kernels and post-kernels which seems to correlate with small Fourier components in the latent pixel space.

The methods section now provides further data on this matter.

Reviewer #2 :

This paper addresses the question of how representations are learned in CNNs. To my mind, this is perhaps the key theoretical problem in understanding CNNs, so it is exciting to see the author's work in this area. This paper takes a statistical physics approach and constitutes a big advance in our understanding of these questions over past work (e.g. Zavatone-Veth et al 2021). My comments are

relatively minor: concerning the presentation of equations in the main text and their relationship to a broader body of work (that does not bring novelty into question).

- *While it is absolutely the author’s choice, I prefer the “Gram matrix” for the pre-activation kernel, and simply “kernel” for the post-activation kernel (see [3]). The logic behind this choice is that we can treat the network as a BNN and a BNN are equivalent to a Gaussian process, with a kernel given by the product of the post-activations with themselves (the author’s free kernel). At the very least, it would be good to have a bit more justification for the choice of these terms. I have to look up which one we’re referring to every single time. Even pre-kernel and post-kernel would be easier to remember.*

Response:

We agree with the referee and adopted his last suggestions (pre-kernels and post-kernels)

- *“Still, the task of finding a useful set of variables that track this flexibility and obey an explicit, general, and closed set of equations — remained unresolved.” There are a couple of nuances here. First, it has been evident for a while through a very large body of work that the right way of describing representations and flexible representation learning in NNs and other deep nonlinear function approximations is going to be through the kernels. For instance, [3] shows that the BNN generative model can be written entirely in terms of just the (free) kernels, without needing to represent the latent kernels, features, or weights. This implies that the (free) kernels are the only necessary set of variables to track flexible representation learning. (Note that [3] differs in that it doesn’t give the closed-form set of equations). Second, [2] gives a GP limit with explicit, general, and closed form sets of equations for understanding representation learning in deep flexible non-linear function approximators. This is not the same result as the present paper, because [2] is restricted to deep GPs, which are closely related to NNs but are not equivalent. That said, [2] was out on arXiv only a couple of months before this article, so it might be okay to ignore it: maybe one for the editor. [2] also has consequences for statements like “GP limits miss out on several qualitative aspects, such as feature learning”. Overall, I’d suggest a formulation something like: “Still, the task of providing an explicit, general closed set of equations describing this flexibility in NNs remains unsolved.”*

Response:

We again adopted the phrasing suggested by the referee and included a reference to these prior works the referee has pointed out

- *“This stands in contrast to the view that data points, in latent space, cluster according to labels (compression of relevant information).” This is very strange. Representation learning in neural networks is known to do*

precisely this: cluster data points according to labels. Indeed, empirically, neural networks usually end up eliminating all information except that relevant to the label. So for instance, for CIFAR-10, you end up with a 10-dimensional top-layer representation. This can be seen empirically, and with various elegant descriptions in linear networks [1]. I think the error is here: “The above also implies that any pre-activation is as likely as any of its rotations in channel space.” There is indeed an elegant rotational symmetry (e.g. [3]), but you have to rotate the whole pre-activation space (i.e. all data points simultaneously). Critically, rotating all data points retains the clustering structure.

Response: The phrasing we used could have indeed been better. In particular, we agree that as it was phrased, it still leaves open the possibility of clustering. The new results section now contains a clearer and hopefully more convincing argument against clustering. One key point is the following counter-example for geometric clustering. Consider n samples with binary labels $y_\mu = \pm 1$ associated with a classification task and $f_c(y_\mu)$ an arbitrary function for each channel, c . In this case, geometric clustering, in the sense that $h_{c\mu} = f_c(y_\mu)$, implies that $C^{-1} \sum_c h_{c\mu} h_{c\nu} = C^{-1} \sum_c f_c(y_\mu) f_c(y_\nu)$. At large C the latter, empirical kernel, is a good proxy to the pre-kernel K . However, its structure only depends on the label and so $K_{\mu\nu} = K_{y_\mu, y_\nu}$, and therefore only takes three values, which we denote as $K_{+,+}, K_{+,-}, K_{-,-}$. Recalling that $y_\mu = \pm 1$, arranging the μ, ν indices into two blocks according to their ± 1 label, and assuming without loss of generality that these two blocks are equal in size— one finds that K is of the block form $[K_{+,+}, K_{+,-}; K_{+,-}, K_{-,-}] \otimes J_{n/2}$ where $J_{n/2}$ is a matrix with all elements equal to one. Hence the empirical kernel is a rank 2 matrix with only two dominant eigenvalues. In contrast, the kernels we find, which describe the experiment very well, have a rich spectrum that does not involve only 2 dominant eigenvalues. Still they embody strong feature learning effects as our Fig. 1. shows.

- *Introduction iii: n is not yet defined? And (assuming that n is the number of data points), the number of data points (squared!) is typically much larger than the number of channels, so wouldn't the emergent scale typically be larger than $O(1)$?*

Response: The training sample's size, n , is now defined at the beginning of the problem statement section. We also added a new figure (Fig. 1 panel c) where we estimate the emergent scale which is indeed often larger than 1. Still, there are two important quantitative points. First, the emergent scale depends on the number of data points squared times the MSE over the number of channels times the input dimension size. The MSE also diminishes with n at large n . In extreme cases (say a linear target and n much larger than the input dimension) it may even decay as $1/n^2$ making the emergent scale constant w.r.t. n . Thus it is difficult to predict the scaling of χ with n . The second point is whether one calculates this MSE using the GP kernel or with the learned data-aware GP kernel. At

moderate feature learning $\chi = O(1)$ this choice does not entail a big change in χ . However, when $\chi \gg 1$, it actually makes a big difference. We opted to define the emergent scale using the GP kernel, as this makes it a more numerically accessible quantity that can be readily tested on CNNs, where solving the EoS is difficult.

Apart from the new figure, these issues are now discussed in more detail in the section describing the numerical results for 3-layer FCN (subsection 2.2) and in the section about the 2-layer CNN (section 2.3).

- *In the main text, results are given for convolutional networks. However,*
 1. *As far as I can see, the theory presented is still novel for fully connected networks (i.e. the authors don't just present an extension of the existing fully-connected theory to CNNs).*
 2. *Results on fully connected networks are much simpler, while still capturing all the key ideas. I'd therefore suggest changing all(?) equations in the main text to fully connected, and leaving the CNN extension to Methods/Supplementary (but obviously keeping the CNN experimental results in the main text).*

Response: Following the comments of both referees on this matter, we now mainly showcase the derivation for the FCN case in the main text and delegate the CNN derivation to the appendix. To complement this, we carried out further numerical experiments on the FCN architecture and report these in the main text.

3. *Eq. 1: The terms in this equation need to be considerably better described:*

- (a) *Can we say explicitly that a , v and w are weights?*

Response: Yes, these are the weights the algorithm finds at equilibrium. This is clarified in the main text.

- (b) *Can we say explicitly that i, j index spatial location?, and what (spatial?) dimensions are of size S_0 , S_1 and N .*

Response: Yes, we now clarified in the main text.

- (c) *Notation for x is a mess: \mathbf{x}_μ is a data point. And at the same time, \mathbf{x}_{i+jS_1} is ... something else? Maybe a patch?*

Response: Yes, \mathbf{x}_{i+jS_1} is a non-overlapping patch of the input.

- (d) *In the main text, can we drop σ_a , σ_v , by taking the weights to be IID Gaussian with variance $1/\text{fan-in}$? Fewer symbols are preferable. At the very least, the σ_a looks like an index and should be σ_a if it is a label.*

Response: Since our results are also applicable to mean-field type scaling, wherein one scales the variances of the weights by the width/number of channels, we opted to keep these more general expressions. We corrected σ_a as suggested. We note that for FCN since we provide the derivation for any number of layers, we

denoted by σ_l^2 the weight decay constant of the l th layer, where now l is a running index.

4. Eq. 4: “The above also suggests that the low-lying spectrum of $Q^{(l)}$ remains relatively inert even in presence of the back-propagation term and tends to align with that of $K^{(l)}$.” isn’t very obvious to me from the presented equation. Is there any way to make this more evident?

Response: In general, given a matrix equation of the type $A^{-1} = B^{-1} + C$ where A, B and C are with bounded real eigenvalues spectrum, we can perform a perturbation theory analysis on the top eigenvalues of A^{-1} (see also Weyl inequality). Let $B^{-1}\phi_\lambda = \lambda\phi_\lambda$, then, leading order in perturbation theory $\lambda \rightarrow \lambda + \phi_\lambda^T C \phi_\lambda$. Thus, eigenvalues with small $\phi_\lambda^T C \phi_\lambda / \lambda$ are weakly perturbed (in the relative sense). Specifically, provided the eigenvalues of C are bounded by a small value, then zero eigenvalues of A ($\lambda \rightarrow \infty$) remain unperturbed in the sense that they remain zero eigenvalues.

The new results section, which includes a discussion of the emergent scale, contains a more detailed account of the differences between $K^{(l)}$ and $Q^{(l)}$ for the specific case of a 3-layer FCN.

5. “Extensions to deeper CNNs and real-world datasets” Slightly misleading given that we’re using a very small subset of CIFAR-10. I’d go with something like “Extensions to deeper CNNs and subsets of real-world datasets”.

Response: The new version of the manuscript now reports on Myrtle-5 experiments with 2048 data points. While still quite small, it is an order of magnitude change from what was previously reported. Still, we adopted the referee’s suggestion.

6. The analysis of the similarity of the kernel matrices is a bit limited. e.g. is it possible to show convergence using e.g. the Frobenius norm as the width increases?

Response: In the new experiments we conducted on FCNs, we show how the empirical kernel’s eigenvalues approach those predicted by the solver. This is shown in Fig. 1 in the main text for the leading eigenvalue of Σ and in Supp. Mat. (6) for the entire spectrum of Σ . In addition, we study how the leading Σ eigenvalue for the 2-layer CNNs approaches its GP value (Supp. Mat. (6)). Absolute Frobenius norm measurements will be somewhat misleading (in a favorable way) since, as evident from these newly reported results, the majority of kernel eigenvalues remain inert.

7. “While the read-out layer typically had an auto-correlation time of the order of 10^3 epochs (at the lowest learning rates) the auto-correlation times for the input layers could be $\sim 10^6$ or larger. To overcome this issue, when analyzing pre-activations of these deeper DNNs we took an ensemble containing 98 and 234 different initialization seeds for the 3-layer CNN and Myrtle-5 respectively.” Ensembling is not a valid solution to slow convergence. The only thing to do

is to run longer or find some other trick to improve convergence (e.g. momentum, tempering the noise?). Maybe that's okay given that it's likely to only push you away from theoretical matching. In general, the procedure for sampling is extremely elaborate, which surprises me. Generally, e.g. SGLD (which is similar, but not the same) is usually very well-behaved. Could you comment on the difficulties?

Response: One of the original goals was to study a training procedure that was as similar as possible to SGD while leaving us with a well-defined partition function, which enables a statistical mechanics analysis. We agree that there are better ways to achieve convergence to equilibrium. The comment in the text may have been misleading - ensembling was done for the simple purpose of parallelizing the gathering of statistics. In Supp. Mat. (6) we now provide further details about the equilibration of our FCNs, looking specifically at the train-loss, test-loss, and largest Σ eigenvalues.

Minor points: Quite a few broken refs. “) is only through the mean and variance of $h(1)$ ” I couldn't find details of the Newton-Krylov method Capitalization in the refs “27. Seeger, M. Pac-bayesian” mutatis mutandis (I am not a fan of Latin phrases).

Response: We addressed these issues in the new manuscript text.

References

- [1] Laurence Aitchison. Why bigger is not always better: on finite and infinite neural networks. In *International Conference on Machine Learning*, pages 156–164. PMLR, 2020.
- [2] Laurence Aitchison. Deep kernel machines and fast solvers for deep kernel machines. *arXiv preprint arXiv:2108.13097*, 2021.
- [3] Laurence Aitchison, Adam Yang, and Sebastian W Ober. Deep kernel processes. In *International Conference on Machine Learning*, pages 130–140. PMLR, 2021.
- [4] Beau Coker, Wessel P Bruinsma, David R Burt, Weiwei Pan, and Finale Doshi-Velez. Wide mean-field bayesian neural networks ignore the data. In *International Conference on Artificial Intelligence and Statistics*, pages 5276–5333. PMLR, 2022.
- [5] Ethan Dyer and Guy Gur-Ari. Asymptotics of wide networks from feynman diagrams. In *International Conference on Learning Representations*, 2020.
- [6] Jiaoyang Huang and Horng-Tzer Yau. Dynamics of deep neural networks and neural tangent hierarchy. In *International conference on machine learning*, pages 4542–4551. PMLR, 2020.

- [7] Jaehoon Lee, Lechao Xiao, Samuel Schoenholz, Yasaman Bahri, Roman Novak, Jascha Sohl-Dickstein, and Jeffrey Pennington. Wide neural networks of any depth evolve as linear models under gradient descent. In H. Wallach, H. Larochelle, A. Beygelzimer, F. d'Alché-Buc, E. Fox, and R. Garnett, editors, *Advances in Neural Information Processing Systems*, volume 32. Curran Associates, Inc., 2019.
- [8] Song Mei, Andrea Montanari, and Phan-Minh Nguyen. A mean field view of the landscape of two-layer neural networks. *Proceedings of the National Academy of Sciences*, 115(33):E7665–E7671, 2018.
- [9] Gadi Naveh, Oded Ben David, Haim Sompolinsky, and Zohar Ringel. Predicting the outputs of finite deep neural networks trained with noisy gradients. *Physical Review E*, 104(6), Dec 2021.
- [10] Gadi Naveh and Zohar Ringel. A self consistent theory of gaussian processes captures feature learning effects in finite cnns. *Advances in Neural Information Processing Systems*, 34, 2021.
- [11] Daniel A Roberts, Sho Yaida, and Boris Hanin. The principles of deep learning theory. *arXiv preprint arXiv:2106.10165*, 2021.
- [12] Ravid Shwartz-Ziv and Naftali Tishby. Opening the black box of deep neural networks via information, 2017.
- [13] Sho Yaida. Non-gaussian processes and neural networks at finite widths. In *Mathematical and Scientific Machine Learning*, pages 165–192. PMLR, 2020.
- [14] Greg Yang and Edward J Hu. Feature learning in infinite-width neural networks. *arXiv preprint arXiv:2011.14522*, 2020.

Reviewer comments, second round -

Reviewer #1 (Remarks to the Author):

I thank the authors for their reply to my comments; their revisions have addressed many of the concerns raised by me and by the other referee. Though the manuscript is much improved, I still have some concerns regarding the framing and presentation of the results, which I detail below.

Major comments:

1. I still find the authors' introduction of their GD+Noise setup to be rather obtuse, particularly as it is only described in words in the main text. I believe that stating that this is discretized unadjusted Langevin sampling from the Bayes posterior would be a clearer description of the setup, particularly for a ML audience. In the rebuttal, the authors argue that they favor the GD+Noise viewpoint because "using this view [they] can compare our result to experiments," but they could just as well have compared their theoretical predictions to the stationary distribution obtained using another sampling algorithm. For example, they could have used Hamiltonian Monte Carlo, as in the large-scale experiments of Izmailov et al., ICML 2021. This is important for applications, as the unadjusted Langevin algorithm is not in practice the method of choice for sampling from BNN posteriors. As a result, I disagree with their assertion that the GD+Noise framing makes their work more accessible to practitioners. Similarly, the authors' comparison of the computational cost of evaluating their theory to that of training a DNN is dependent on their choice of sampling algorithm. Performing approximate maximum-likelihood inference using SGD would be far faster than full posterior sampling.
2. The main text does not clearly describe the effect of activation function either on feature learning or on the tractability of the mean field theory. It could be useful to provide some more discussion of this issue.
3. The main text focuses on training set predictions. This should be made more clear, and the proposed method to study test set statistics should be stated in its own subsection of the main text, rather than being deferred to the supplement. This caveat should also be stated along with the claim "Though bounds have been derived, we are not aware of any analytical predictions for the performance of finite non-linear 2-layer DNNs, let alone CNNs. It is therefore a natural first application of our approach" in Section 2.3.
4. Though the readability of the revised manuscript is significantly better than the initial submission, there is still room for improvement. The equations for the log-posterior and the weight dynamics should be given explicitly; introducing these vital elements of the problem setup only in words is not enough. Moreover, the discussion of related work in the last paragraph of the introduction is not entirely satisfactory, as it does not make clear what gaps in that body of work the present manuscript fills.

Minor comments:

The addition of quantile-quantile plots is useful, but the legibility of these figures could be improved by making the axis scales equal.

I don't think the authors' description of work by Aitchison et al. on deep kernel processes (ref. 19 in the updated manuscript) as "phenomenological" is accurate.

The authors should proofread the citation list, as the arXiv versions of several published works are referenced (e.g., Zhang et al 2016), and duplicates remain in the supplementary material.

I think Equation 9 and Figure 3 are likely to confuse the reader without the addition of further expository prose; they could be deferred to the methods or to the supplemental material

Italicizing pre-kernel and post-kernel is distracting and unnecessary.

Consider using a symbol other than δ to represent the discrepancy between the target and the average predictor.

Response to Reviewer’s Comments

We would like to thank the reviewer for his/her constructive comments, which helped us improve our manuscript significantly. We have revised the manuscript to incorporate all the comments received. In particular, we added detailed analysis on the generalization of our theory to test points. We now introduce our work putting more emphasis and motivation on the Bayesian framework, and also shortly discuss the merits of such a framework. Last, we clarify in more detail what gaps in the literature are addressed by our work.

In what follows, we explain how each and every comment was addressed, with references to parts of the paper that have been modified pursuant to the reviewer’s suggestions. Our responses appear in the non-italic font.

Reviewer #1 :

I thank the authors for their reply to my comments; their revisions have addressed many of the concerns raised by me and by the other referee. Though the manuscript is much improved, I still have some concerns regarding the framing and presentation of the results, which I detail below.

Major comments:

1. *I still find the authors’ introduction of their GD+Noise setup to be rather obtuse, particularly as it is only described in words in the main text. I believe that stating that this is discretized unadjusted Langevin sampling from the Bayes posterior would be a clearer description of the setup, particularly for a ML audience. In the rebuttal, the authors argue that they favor the GD+Noise viewpoint because "using this view [they] can compare our result to experiments," but they could just as well have compared their theoretical predictions to the stationary distribution obtained using another sampling algorithm. For example, they could have used Hamiltonian Monte Carlo, as in the large-scale experiments of Izmailov et al., ICML 2021. This is important for applications, as the unadjusted Langevin algorithm is not in practice the method of choice for sampling from BNN posteriors. As a result, I disagree with their assertion that the GD+Noise framing makes their work more accessible to practitioners.*

Response: We thank the reviewer once more for their detailed comments. We agree that the GD+noise description in the previous version was not clear enough and did not tie well with the literature on the topic. Accordingly, we added a more detailed description in the Methods section and

mentioned that this is indeed the discretized Unadjusted Langevin Algorithm (ULA) for sampling from the Bayes posterior, as suggested. We also added some citations on this point: [1, 2, 3], which point to rather good convergence properties of ULA. While this method of posterior sampling may be slower to converge compared to other methods (such as HMC[4]), it is simpler (e.g. has no Metropolis acceptance step) and admits an intuitive comparison with vanilla SGD, where hyperparameters such as learning rate and weight decay and noise level (or mini-batch size for SGD) can be experimented with and compared across these different training protocols.

In addition, following the referee’s advice our problem-setting section now focuses more on the Bayesian viewpoint and details all the relevant quantities, in particular, the log-posterior in terms of just the network’s output (as opposed to the log-posterior of the outputs and pre-activations which appeared in the last version of the manuscript).

Similarly, the authors’ comparison of the computational cost of evaluating their theory to that of training a DNN is dependent on their choice of sampling algorithm. Performing approximate maximum-likelihood inference using SGD would be far faster than full posterior sampling.

Response: We completely agree that the time-complexity aspects of our EoS solver, especially compared with the various possible DNN optimization schemes, is a matter which requires further clarification. However, we do not feel the current manuscript is the place to do so. Making a disciplined comparison of this type requires extensive numerical experiments as well as working out various, potentially simple ways, in which our EoS solver could be improved. Indeed, similar problems of inverting large matrices have been encountered in the field of GPs and various smart tools have been developed. As we said before, the main purpose of this work is not to offer a new algorithm to replace BNN samplers - rather, it is to provide analytical insights. The emergent scale, our analytical solution for the 2-layer CNN, our identification of slow variables, and the effective GP description all testify in favour of this.

2. *The main text does not clearly describe the effect of activation function either on feature learning or on the tractability of the mean-field theory. It could be useful to provide some more discussion of this issue.*

Response: The revised main text now extends the previous short comments we had on ReLU activations. Specifically, when discussing our numerical experiment on Myrtle-5 we comment again that our EoS generalizes straightforwardly to ReLU, we refer to Supp. Mat. 5 which derives this, and mention that this entails some 1d integrals that need to be obtained numerically. We further comment, that in this case additional variables need to be tracked— the means of pre-activations. Notwithstanding, the EoS are just as tractable as before (in fact, as a part of an on-going project we are working with an EoS solver for ReLU). This raises an in-

interesting question of how much feature learning now goes into changes to kernels versus changes to means. This matter needs further investigation, as do other questions regarding the role of activation functions. The revised version points that our Myrtle-5 experiment supports the fact that changes to kernels still play a dominant role. Further research into those questions is left for future work.

3. *The main text focuses on training set predictions. This should be made more clear, and the proposed method to study test set statistics should be stated in its own subsection of the main text, rather than being deferred to the supplement. This caveat should also be stated along with the claim "Though bounds have been derived, we are not aware of any analytical predictions for the performance of finite non-linear 2-layer DNNs, let alone CNNs. It is therefore a natural first application of our approach" in Section 2.3.*

Response: While we have indeed placed more emphasis on train-set results, we do not believe our work contains any major caveats here. The original submission already contained concrete predictions on test points (Fig. 2. of the previous manuscript) in the context of non-linear 2-layer CNN. The first revised manuscript contained a Supp. Mat. Section 1.4.1. which provided a concrete way of generalizing the EoS of any DNN to test point by formally treating the test point as an additional training point with its own "noise" parameter and then sending it to infinity, thereby obtaining EoS which include the test point.

We agree that this matter hasn't been presented clearly enough. To improve this, the main text now details how the EoS of the two-layer CNN generalize to test points and provides further derivation details in Supp. Mat. 1.4.1. Specifically, we perform the aforementioned limit explicitly on the EoS. We also comment on generalizations to test points in the general context of section 2.1.

4. *Though the readability of the revised manuscript is significantly better than the initial submission, there is still room for improvement. The equations for the log-posterior and the weight dynamics should be given explicitly; introducing these vital elements of the problem setup only in words is not enough. Moreover, the discussion of related work in the last paragraph of the introduction is not entirely satisfactory, as it does not make clear what gaps in that body of work the present manuscript fills.*

Response: The revised introduction now contains further clarifications concerning gaps in the literature. In particular, the fact that in the feature learning regime, no concrete set of equations existed to describe learning in anything but the simplest non-linear DNNs having one trainable layer. Obviously, concrete is somewhat subjective, and so we stress that we mean a set of equations involving only a finite set of basic matrix manipulation. Taking a view that analytical advancement in complex systems are highly correlated with identifying slow variables, we further point out that, away

from the GP-limit, no subset of variables was known other than the entire set of DNN weights. While it is true that people have studied kernel flexibility as an important degree of freedom, our equations of state (EoS) show that kernel flexibility (and “mean flexibility” for antisymmetric activations) form a complete set of variables in the appropriate parameter region as also manifested in our numerical results— and no further information on the weights is needed. Last, to demonstrate the analytical benefits of our EoS in the most crisp manner, we emphasize that our EoS facilitate the derivation of what we believe is the first prediction of the performance of non-linear CNN with more than one trainable layer.

Last, We also added a more detailed description of the weight dynamics in the Methods section, and clarification and definition of the log posterior in the problem statement.

Minor comments:

1. *The addition of quantile-quantile plots is useful, but the legibility of these figures could be improved by making the axis scales equal.*

Response: The quantile-quantile plots are used here to quantify signatures of non-Gaussianity namely, deviations from the $y = x$ line. If we put all panels on the same scale (say $[-4, 4]$ for both x and y axes), then we would largely miss the strong deviation seen in the first and second columns of the second row. Consequently, we feel such scaling (if we understood the referee correctly) would look misleadingly favorable to our claims.

2. *I don't think the authors' description of work by Aitchison et al. on deep kernel processes (ref. 19 in the updated manuscript) as "phenomenological" is accurate.*

Response: We changed this citation to “While quantitatively different, our approach is similar in spirit to the layer-wise Gaussian Processes algorithm [19] inspired by DNN experiments. However, our approach provides a more accurate first-principles description of trained DNNs.

3. *The authors should proofread the citation list, as the arXiv versions of several published works are referenced (e.g., Zhang et al. 2016), and duplicates remain in the supplementary material.*

Response: We replaced all suitable references.

4. *I think Equation 9 and Figure 3 are likely to confuse the reader without the addition of further expository prose; they could be deferred to the methods or to the supplemental material*

Response: We delegated much of the details of the 3-layer CNN to the Methods section. In particular, Eq. 9. and half of figure 3. The other half which sketches the general structure of our theory was moved up closer to the results sections.

5. *Italicizing pre-kernel and post-kernel is distracting and unnecessary.*

Response: We removed the italic font.

6. *Consider using a symbol other than δ to represent the discrepancy between the target and the average predictor.*

Response: We replace the symbol δ by ε .

References

- [1] Vempala, S. & Wibisono, A. Rapid convergence of the unadjusted langevin algorithm: Isoperimetry suffices. *Advances in neural information processing systems* **32** (2019).
- [2] Durmus, A. & Moulines, E. Nonasymptotic convergence analysis for the unadjusted langevin algorithm. *The Annals of Applied Probability* **27**, 1551–1587 (2017).
- [3] Mou, W., Flammarion, N., Wainwright, M. J. & Bartlett, P. L. Improved bounds for discretization of langevin diffusions: Near-optimal rates without convexity. *Bernoulli* **28**, 1577–1601 (2022).
- [4] Izmailov, P., Vikram, S., Hoffman, M. D. & Wilson, A. G. G. What are bayesian neural network posteriors really like? In *International conference on machine learning*, 4629–4640 (PMLR, 2021).

Reviewer comments, third round -

Reviewer #1 (Remarks to the Author):

I thank the reviewers for their responses. I do not have any other comments.